# Genetic variation at transcription factor binding sites largely explains phenotypic heritability in maize

Julia Engelhorn [1,2,3], Samantha J. Snodgrass [4], Amelie Kok [1,2,5,6], Arun S. Seetharam [4], Michael Schneider [6], Tatjana Kiwit[1,2], Ayush Singh [7], Michael Banf[8], Duong Thi Hai Doan[1,2], Merritt Khaipho-Burch[9], Daniel E. Runcie [10], Victor A. Sánchez-Camargo [11], Rechien Bader [11], J. Vladimir Torres-Rodriguez[12], Guangchao Sun [13], Maike Stam [11], Fabio Fiorani [14], Sebastian Beier [15], James C. Schnable [12], Hank W. Bass [7], Matthew B. Hufford [4], Benjamin Stich [5,6,16], Wolf B. Frommer [1,2,5,17], Jeffrey Ross-Ibarra [18] & Thomas Hartwig [1,2,5] ✉

Comprehensive maps of functional variation at transcription factor (TF) binding sites (*cis*-elements) are crucial for elucidating how genotype shapes phenotype. Here, we report the construction of a pan-cistrome of the maize leaf under well-watered and drought conditions. We quantified haplotype-specific TF footprints across a pan-genome of 25 maize hybrids and mapped over 200,000 variants, genetic, epigenetic, or both (termed binding quantitative trait loci (bQTL)), linked to *cis*-element occupancy. Three lines of evidence support the functional significance of bQTL: (1) coincidence with causative loci that regulate traits, including *vgt1*, *ZmTRE1* and the MITE transposon near *ZmNAC111* under drought; (2) bQTL allelic bias is shared between inbred parents and matches chromatin immunoprecipitation sequencing results; and (3) partitioning genetic variation across genomic regions demonstrates that bQTL capture the majority of heritable trait variation across ~72% of 143 phenotypes. Our study provides an auspicious approach to make functional *cis*-variation accessible at scale for genetic studies and targeted engineering of complex traits.

Over the past two decades, genome-wide association studies (GWAS) have transformed our understanding of the inheritance of many complex traits in important crops such as maize. Several studies have estimated that non-coding variation accounts for about 50% of the additive genetic variance underlying phenotypic diversity in plants[1–4]. Although identification of functional non-coding variants is advancing with the development of new genomics technologies[5], it remains challenging to discern functional variants that impact *cis*-elements efficiently and at cistrome (defined as the genome-wide set of *cis*-acting regulatory loci) scale. Knowing which loci to target has become one of the obstacles for

trait improvement by targeted genome editing[5–7]. Scalable methods to construct comprehensive *cis*-element maps are essential to understand complex transcriptional networks that underlie development, growth and disease. The potential of *cis*-element maps has been demonstrated by the ENCODE projects that exist for many eukaryotes, including humans. However, genome-wide, high-resolution maps of functional variants are currently lacking in plants[8]. Despite many successes, GWAS generally suffer from insufficient resolution, which limits the identification of individual causal single-nucleotide polymorphisms (SNPs) or insertions or deletions (INDELs) and cannot provide independent

molecular information on the potential function of variants, requiring laborious follow-up analyses of numerous individual loci[7].

An alternative approach to identify functional polymorphisms would be to annotate non-coding variants within a GWAS region based on their association with TF binding. This approach has considerable potential, as TF activity has an important role in the regulation of genes, and thereby traits, and the affinity of TF binding is mostly determined by specific local sequences (*cis*-elements)[9,10]. Identifying *cis*-elements for individual TFs through approaches such as chromatin immunoprecipitation with sequencing (ChIP–seq) is time-consuming, not strictly quantitative, limited in scope and often provides relatively low resolution of functional regions. By contrast, MNase-defined cistrome occupancy analysis (MOA-seq) identifies putative TF binding sites globally, in a single experiment with relatively high resolution and yields footprint regions typically of <100 bp (ref. 11). In maize, MOA-seq identified ~100,000 TF-occupied loci, including about 70% of the sequences (bp overlap) identified in more than 100 ChIP–seq experiments[11,12]. Notably, many of the MOA footprint regions were previously uncharacterized, with only 35% identified in previous assay for transposase-accessible chromatin sequencing (ATAC-seq) data; by contrast, MOA-seq identified 76% of previous ATAC-seq peaks[11]. Similarly, an analysis of small MNase-defined fragments from *Arabidopsis* seedlings revealed more than 15,000 accessible chromatin regions missed by ATAC-seq or DNase-seq[13].

Here, we quantified haplotype-specific TF footprints across the maize pan-genome with MOA-seq, using F1 hybrids that share a common reference to minimize biological, technical and *trans*-effect variation between the haplotypes. We defined a maize leaf pan-cistrome and identified ~210,000 variants that were genetic, epigenetic, or both linked to haplotype-specific variation in MOA coverage at candidate *cis*-element loci, which we term bQTL. The bQTL explained the majority of heritable trait variation in >70% of the tested traits in the nested association mapping (NAM) panel. Haplotype-specific TF footprints coincided with causative loci known to affect leaf angle, branching and flowering time traits, and identified *ZmTINY* (Zm00001eb120590) and more than 3,500 drought-response putative *cis*-regulatory regions as candidate loci for future smart breeding.

## Results

### Quantification of functional *cis*-variation

To focus on genetic differences affecting TF binding in *cis*, we quantified TF footprints (defined as the area significantly covered by MOA-seq reads) specific to each haplotype in F1 hybrids with a shared reference parent (B73) (Fig. 1a). We applied MOA-seq to nuclei of the inbred lines B73 (ref. 14) and Mo17—founders of key maize breeding populations whose hybrid has been extensively studied[15–17]—and their F1 hybrids. MOA footprints were determined by mapping sequencing reads to a concatenated hybrid genome and retaining reads that mapped uniquely (Supplementary Table 1; for some analyses we used reads mapping equally well to two locations; Methods). We detected 327,029 MOA footprints or peaks (false discovery rate (FDR) of 5%) with strong correlation across biological replicates (Pearson's correlation coefficient > 0.95; Supplementary Fig. 1). A total of 53,220 genes in the F1, representing 67.9% of B73 and Mo17 annotated genes (5 kb upstream and 1 kb downstream; Supplementary Table 2) were flanked by at least one MOA footprint. Furthermore, the MOA footprints harbored 325,933 SNPs, which we termed MOA polymorphisms (MPs). Among all MPs, we identified 48,505 with an allelic ratio that significantly deviated from the expected 1:1 in F1s, which we termed allele-specific MPs (AMPs; binomial test with 1% FDR, validated with whole-genome sequencing (WGS) controls; Supplementary Fig. 2).

The vast majority (88% or 194,594 out of 221,187) of all MPs showed no significant difference in their allelic bias comparing F1 (B73 and Mo17 haplotypes) to B73 versus Mo17 inbred alleles (black dots in Fig. 1b), and about 90% (31,949 out of 35,638) of AMP sites in the B73 × Mo17 F1

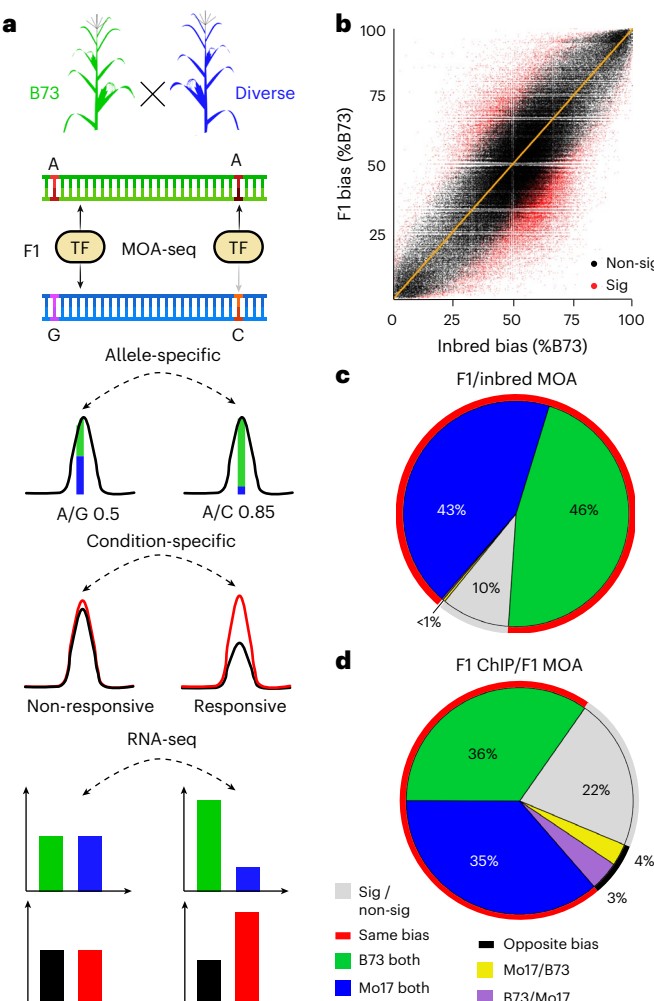

**Fig. 1 | Quantitative *cis*-element occupancy analysis in F1 hybrids.**
**a**, Haplotype-specific MOA flowchart: 1) Nuclei purified from diverse nested (B73 common mother) F1s are analyzed by MOA-seq, producing small, non-nucleosomal, protein–DNA interaction footprints. 2) SNPs in MOA peaks (MPs) allow the identification, quantification and, in a population, association of variants coupled to occupancy of putative *cis*-elements. Allele-specific MOA footprints can be compared between treatments; for example, well-watered versus drought. 3) Allele-specific mRNA-seq allows further characterization of functional variants associated with gene regulation. Created with Biorender.com **b**, Correlation of haplotype-specific MOA-seq data at all MPs in nuclei from B73 versus Mo17 inbreds (*x* axis) versus those from the F1 (*y* axis) (Pearson correlation, 0.78). MPs with significant (red, *P* < 0.05, expected *trans*) and without significant (black, expected *cis*) differences between F1 and parental alleles are marked. **c**, Genome-wide comparison of allelic bias (50–60% to one allele considered no bias, >60% considered biased) at B73 × Mo17 F1 AMP sites to inbred B73 versus Mo17 data. Only sites that displayed binding in both inbreds and hybrids were considered. **d**, Genome-wide directionality analysis, comparing AMPs detected by MOA-seq to ChIP–seq data of a single TF BZR1 (ref. 17) in the B73 × Mo17 hybrid. Only sites that displayed binding in both ChIP and hybrid MOA were considered. In **c** and **d**, MOA occupancy was largely consistent (red circle) between either F1 and parents or compared to ChIP–seq, respectively, in showing bias towards B73 (green) or Mo17 (blue) in both cases, with a smaller fraction of allele-specific F1 MOA sites showing no bias (gray) in inbreds or ChIP–seq, or bias to the opposite parent or allele (B73 in F1 and Mo17 in inbred or ChIP, purple, or Mo17 in F1 and B73 in inbred or ChIP, yellow).

showed bias towards the same allele as when comparing the inbred parents (red line in Fig. 1c). Fewer than 0.6% (199 out of 35,638) of AMP sites showed bias in the opposite direction. In the F1, allele-specific bias at AMPs should not be affected by *trans*-factors, biological or technical variation, as the relative haplotype differences originate from the same

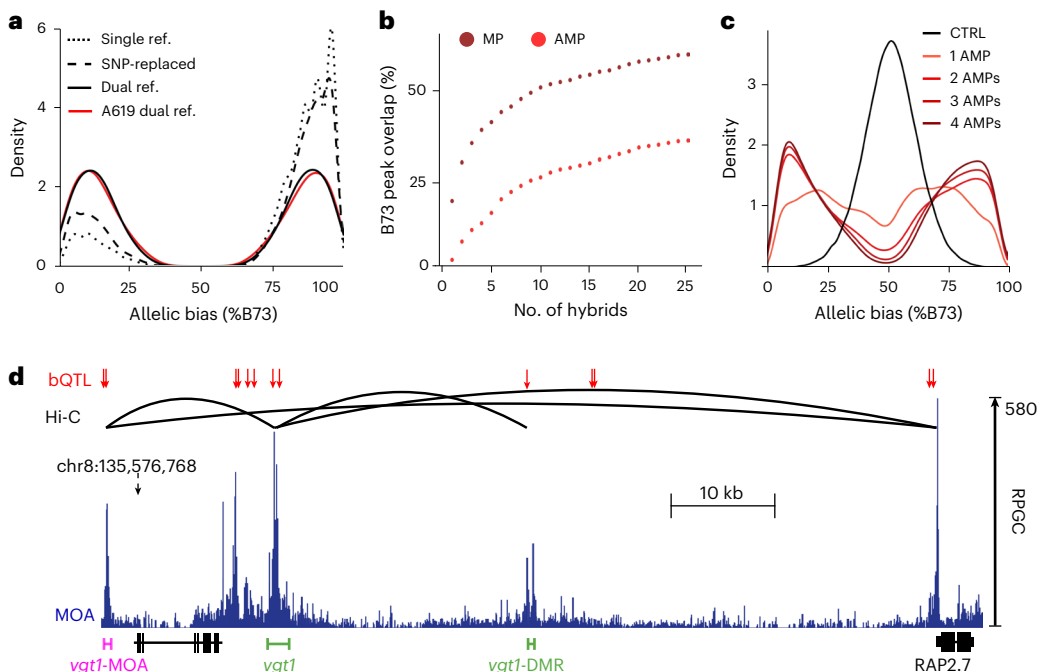

**Fig. 2 | Construction of a maize leaf pan-cistrome. a**, Mapping strategies comparison showing the density of AMPs (allele-specific occupied sites) over the percentage of binding to B73. B73 × HP301 F1 MOA data were analyzed using either only B73 as reference genome (single ref.), a pseudo-genome with B73/HP301 SNPs replaced by Ns (SNP-replaced) or our dual-parent mapping strategy using a concatenated B73 × HP301 genome (dual ref.). Without mapping bias, a symmetric distribution is expected (as observed for dual ref.), while a higher density at higher B73 allelic bias indicates biased mapping to the reference genome B73 (single ref. and SNP-replaced). For the A619 F1 (no assembled genome available), our 'reference-guided' strategy (see Methods for details) showed similar AMP-balanced haplotypes without reference bias

(A619 dual ref.). **b**, Additive percent of B73 MOA peak covered by MPs (brown) and AMPs (red) relative to the number of F1s analyzed. **c**, Density of mean MOA binding frequencies over all F1s carrying a SNP at positions where at least one, two, three or four F1s had AMPs, compared to a control with randomized binding frequencies. **d**, Overview of bQTL (red arrows), MOA coverage (blue) and Hi-C interaction sites (black lines, Hi-C from a previous publication[32]) near the classical flowering repressor *RAP2.7*. bQTL overlap with both known enhancers, *vgt1* and *vgt1*-DMR (green), associated with *RAP2.7* expression. An additional bQTL, termed *vgt1*-MOA (magenta), also interacts with *vgt1* and the RAP2-7 promoter.

F1 cells. The high concordance between haplotype-specific bias in the F1 and inbreds at AMP loci is consistent with this expectation, further establishes the reproducibility of the assay and indicates that the majority of AMPs are coupled to genotypic differences in *cis* at the binding site, rather than resulting from *trans*-acting or *cis*-by-*trans* interaction effects. The fact that some differences are observed, however, underpins the importance of using F1 hybrids rather than inbred lines, in which *trans*-acting and *cis*-acting effects cannot be easily disentangled.

To independently validate haplotype-specific, MOA-defined, putative TF footprints in B73 × Mo17, we compared AMPs to recently published allele-specific ChIP–seq data of the major brassinosteroid TF ZmBZR1 in the same F1 (ref. 17). More than 70% of AMPs overlapping with ZmBZR1 binding sites showed allelic bias in the same direction in both studies (red line in Fig. 1d). About 22% of AMPs showed no bias in the ChIP–seq data, probably because of the lower resolution of haplotype-specific ChIP–seq (~500 bp fragments compared to ~65 bp for MOA-seq). Only 7% of AMPs showed bias for different alleles than in ChIP–seq, potentially reflecting biological differences in the tissues analyzed (meristem and leaf versus leaf) or ectopic BZR1 activity owing to exogenous brassinosteroid treatment[17]. Detailed comparison of MOA-seq occupancy to the ZmBZR1 ChIP–seq data demonstrates the accuracy and resolution of our approach, accurately predicting expression of downstream genes and enabling the identification of likely causal polymorphisms within the TF binding site (Extended Data Fig. 1 and Supplementary Note 1). Together, these examples illustrate the potential of MOA-seq to annotate candidate *cis*-regulatory elements with quantitative chromatin footprint data that connects *cis*-variation to biases in *cis*-element occupancy.

## Defining functional sites in a maize pan-cistrome

To define a leaf pan-cistrome of maize, we analyzed a population of 25 F1 hybrids using haplotype-specific MOA-seq (Fig. 1a). The hybrid population, created by crossing 25 inbred lines with high-quality genome assemblies[18–21] to the reference genome line B73, represents a diverse set of maize including many of the parents of an important mapping population and several important genetic stocks (Supplementary Table 3). We analyzed allele-specific TF occupancy and mRNA abundance in leaf blades of each F1 cross (Supplementary Tables 1 and 4). By aligning MOA-seq and RNA-seq reads to concatenated dual-reference genomes rather than a single reference, our approach resolves issues of reference bias that confound most allele-specific analyses[22] (Fig. 2a; Methods). We identified an average of 237,000 MOA peaks (FDR, 5%) per F1, covering approximately 2% (around 80 Mbp) of each hybrid genome (Supplementary Fig. 4). On average, 19.9% (14–30%) of MPs showed a significant allelic bias (binomial test, FDR, 1%; Supplementary Table 5) with an overall even split between the parental alleles (50.2% B73 and 49.8% diverse parents; Supplementary Fig. 5 and Supplementary Table 5). It is noteworthy that the average rate of AMPs (19.9%) closely matches allele-specific TF binding sites detected by the gold standard of ChIP–seq for an individual TF (18.3%)[17]. In total, AMPs overlapped with 35.6% of all MOA footprint peaks in B73 (Fig. 2b), and plots of the identified MOA peaks and cumulative base pairs indicate that our diverse population is near saturation and has identified the majority of the B73 leaf cistrome (Supplementary Fig. 6).

We next sought to identify variants, genetic, epigenetic, or both, associated with differences in MOA-detected TF occupancy between haplotypes, or bQTL, in our population. We first verified that F1s that

shared haplotypes at AMP loci also share similar patterns of allelic bias (Fig. 2c), indicating that our SNPs were in sufficient linkage disequilibrium with causal differences to perform association analysis. Differences in DNA methylation between parental alleles can affect TF binding affinity[17,23]. After validating that DNA methylation differences at AMPs detected between F1 haplotypes were consistent between the parental lines (Extended Data Fig. 2a,b), we added previously published methylation data for 24 of our parental lines[18–20]. We performed linear modeling to test all positions that are MPs (SNPs or INDELs in MOA peaks; Methods) in at least two lines for association of MOA signal variation with either the genotype information, DNA methylation level or both. We identified a total of 176,613 (147,942 SNP and 28,671 INDEL loci, FDR < 0.05; Supplementary Table 6) significant associations, termed bQTL, of which 93,682, 51,192 and 31,739 bQTL were a result of genotype variation alone, DNA methylation variation alone or both features, respectively (Extended Data Fig. 2c,d). As expected, the genome-wide distribution of bQTL was distinct from all SNPs and more closely matched those of previously published allele-specific TF binding sites determined by ChIP–seq[17] (Supplementary Fig. 7). A notable bQTL includes an 8 bp (T/TTAGCGTGT) INDEL in the hypervariable region of the *ZmBIF2* (Zm00001eb031760) promoter (Supplementary Note 1) at a site bound by multiple TF families[12] (bZIP, EREB, bHLH, MYB and WRKY; Extended Data Fig. 3). Overall, we found that INDEL variants show patterns very similar to SNPs, with, for example, 63% of INDEL bQTL overlapping with a SNP bQTL within 65 bp. We thus focused our further analysis on the SNP bQTL while providing the INDEL bQTL to increase the resolution of the pan-cistrome map.

### bQTL coincide with known, causative regulatory loci

Detailed analyses of regulatory variation for a number of maize genes provide an opportunity to compare bQTL to previously identified causal variation. One bQTL was directly adjacent to the YABBY TF binding site underlying the leaf architecture QTL upright plant architecture2 (ref. 24) (Zm00001eb073010) (Extended Data Fig. 3). bQTL also identified haplotype-specific footprints at flowering time loci, including the causative transposon insertions at *ZmCCT9* (Zm00001eb391230) that was targeted by selection during maize adaptation to higher latitudes[25,26], INDEL-2339 in the promoter of the FT-like *ZmZCN8* (ref. 27) (Zm00001eb353250), a 850 bp structural variant in the promoter of *ZmPHYB2* (ref. 28) (Zm00001eb396030) as well as multiple GWAS hits for flowering time (Extended Data Fig. 3). In addition to identifying bQTL in both of the known distal regulatory regions, vegetative to generative transition 1 (*vgt1*) and *vgt1*-DMR, of the key flowering time locus *ZmRAP2.7* (refs. 29–31) (Zm00001eb355240), our bQTL analysis identified an undescribed, third putative enhancer more than 100 kb upstream, which we termed *vgt1*-MOA (Fig. 2d). Hi-C long-range interaction data[32] confirmed that *vgt1*-MOA physically interacts with both *vgt1* and the proximal *ZmRAP2.7* promoter (Fig. 2d). However, future functional tests are needed to establish whether *vgt1*-MOA effects *ZmRAP2.7* expression alone or in combination with *vgt1* and/or *vgt1*-DMR.

We further observed that bQTL colocalized with the regulatory variation upstream of *ZmGT1* (Zm00001eb007950), which is targeted by ZmTB1 (Zm00001eb054440), with the two forming a regulatory module involved in bud dormancy and growth repression[33]. bQTL coincide with the transposon-associated causal regulatory region for prolificacy (*prol1.1*) upstream of *ZmGT1*, including one bQTL directly adjacent to the TB1 binding site[34] (Supplementary Fig. 8).

Our MOA-seq pan-cistrome also provides an opportunity to evaluate how variation at these sites compares to changes in *cis*-element occupancy. For example, an INDEL in the TREHALASE1 (*ZmTRE1*, Zm00001eb021270) promoter has been associated with both trehalose amounts and *ZmTRE1* transcript levels in maize[35]. We observed haplotype-specific footprints, both at a previously reported 8 bp insertion[35] and an additional SNP 29 bp upstream, which coincided with a bQTL (Supplementary Fig. 9). Notably, although the 8 bp insertion

creates a potential ABI motif (TGCCACAC), the *ZmTRE1*-bQTL overlaps with a DOF binding motif (AAAAGGTG). Previously published ChIP–seq results confirm that the *ZmTRE1*-bQTL site is targeted by ZmDOF17 (ref. 12) (Supplementary Fig. 9). Furthermore, all alleles (6 out of 6) in our F1 population without the 8 bp insertion and with the non-canonical DOF motif (C instead of G) at the bQTL site showed concomitant low MOA signal (strong bias towards B73's G allele) and *ZmTRE1* mRNA levels (higher B73 mRNA level) (Fig. 3a,b). In another example, ZmSUBTILISIN11 (*ZmSUB11*, Zm00001eb152020) has been associated with cell wall compositions, peduncle vascular traits and abscisic acid (ABA) levels[36,37]. A previously identified *cis*-expression QTL lead SNP for *ZmSUB11* transcript levels[38] coincided with a bQTL in its proximal promoter, and we observed a strong correlation of haplotype-specific MOA footprints at the bQTL and *ZmSUB11* transcript levels (Fig. 3a,b).

### MOA bQTL correlate with transcript levels

If variation in MOA coverage accurately captures TF binding affinity, we would expect to see associations between haplotype-specific MOA coverage and transcript abundance in our F1s. Indeed, we find that the promoters (within 3 kb upstream of the transcription start site (TSS)) of genes with significant allele-specific expression (ASE, *P* < 0.05; Methods) were ~34% and ~74% enriched for the presence of AMPs compared to all expressed and non-haplotype-specific (non-ASE, *P* > 0.95) expressed genes, respectively (Fig. 3c, Supplementary Table 7a–c and Supplementary Fig. 10). Genotype-associated bQTL were also substantially more likely to be in high linkage disequilibrium (>0.6) with nearby *cis*-expression QTL in a panel of 340 maize genotypes[38] than matched background SNPs (bgSNPs) (49.1% more intergenic bQTL relative to bgSNPs (456/306) and 28.2% more total bQTL versus bgSNPs (1,775/1,384), respectively) (Fig. 3d). These broad patterns are reflected at the level of individual genes as well. For example, all of the NAM parents showing greater MOA occupancy at the bQTL upstream of PHOSPHOGLYCERATE MUTASE1 (*ZmPGM1*, Zm00001eb196320) showed significantly increased abundance of the NAM transcript, whereas F1s with no polymorphism between B73 and NAM in their promoter or 5′ untranslated region showed no significant difference in haplotype-specific transcript levels (Fig. 3e,f). Two NAM parents, Ki3 and CML69, showed much lower *PGM1* transcript levels (Fig. 3f), while no significant variation in MOA footprint was detected. Instead, Ki3 and CML69 harbored a PIF/Harbinger transposon insertion accompanied by hypermethylation between the MOA peak and *PGM1* TSS, not found in any of the haplotypes (including B73) with higher *PGM1* transcript levels (Fig. 3e,f and Supplementary Fig. 11).

### Variation in DNA methylation can predict MOA occupancy

The vast majority of TFs in *Arabidopsis* have been shown, in vitro, to have higher binding affinity to hypomethylated DNA[23]. We explored this association in our data, focusing on variation in CG and CHG methylation (mCG/mCHG), as they accounted for >99.8% of methylation differences at MOA sites. DNA methylation differences (following a previous publication[39], one allele <10% methylated and the other >70%) overlapped with 14.8% of MPs in the F1s. At AMPs, haplotype-specific mCG/mCHG overlap increased by 2.6-fold (38.1%) and reached more than half (51.5%) for AMPs with a strong haplotype-bias (≥85% to one allele) (Fig. 4a). We observed a very strong correlation between a higher footprint occupancy and the hypomethylated allele (Fig. 4b), with 98.2% of AMPs showing higher MOA coverage at the hypomethylated allele. Furthermore, nearly half of the remaining 1.8% AMPs biased towards hypermethylation alleles did not display methylation difference immediately surrounding the AMP (11 bp window), intimating that there may be no actual methylation difference at the occupied site despite hypermethylation of the surrounding region (41 bp window) (Supplementary Fig. 12). On average, the vast majority (71.2%) of F1s that shared differentially methylated alleles at a given locus also

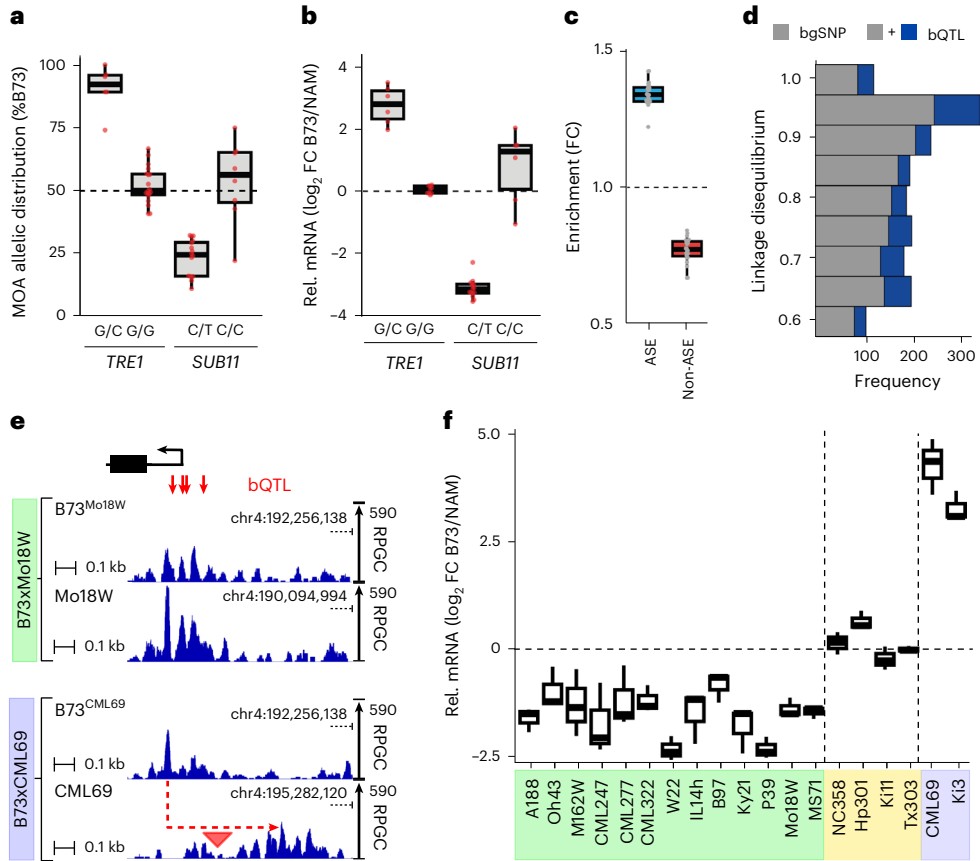

**Fig. 3 | bQTL correlate with haplotype-specific transcript variation. a**, Allelic distribution (%B73) of MOA reads at bQTL in either the *TRE1* (Zm00001eb021270, bQTL: B73-chr1:80,826,022) or *SUB11* (Zm00001eb152020, bQTL B73-chr3:198,733,446) proximal promoters (*TRE1*: 19 F1s with G/G and six F1s with G/C alleles; *SUB11*: eight F1s with C/C and 16 F1s with C/T alleles). **b**, Haplotype-specific mRNA counts for *TRE1* and *SUB11* grouped by their respective bQTL alleles in **a** (*TRE1*: nine F1s with G/G and six F1s carrying G/C alleles; *SUB11*: six F1s with C/C and 15 F1s with C/T alleles). Only lines with polymorphic alleles were considered. FC, fold change. **c**, Genes with ASE and non-ASE mRNA abundance are significantly more and less enriched for AMPs in their 3 kb upstream promoter, respectively (*n* = 24 F1s, hypergeometric test; Supplementary Table 7a–c). **d**, bQTL are more often in linkage disequilibrium with *cis*-expression QTL, identified in roots of 340 recombinant inbred lines than matched bgSNPs. **e**, Average, normalized MOA coverage for B73 and NAM alleles of B73 × Mo18W

and B73 × CML69 upstream of *PGM1*, Zm00001eb196320. The Mo18W allele showed significantly higher MOA occupancy (green panel), while the CML69 allele showed similar MOA coverage to B73, yet peaks were shifted ~300 bp because of a MITE transposon (red triangle) insertion (purple panel). **f**, Allele-specific mRNA counts (*n* = 3 biological replicates) of *PGM1* in the different F1 hybrids. Colors indicate MOA ratios at bQTL: NAM > B73 (green, >60% bias to NAM, non-B73 allele for at least one bQTL), B73 = NAM (yellow, %B73 occupancy between 40% and 60%, sharing B73 genotype) and transposon insertion haplotype (purple). All F1s within the NAM > B73 and transposon category displayed significantly higher or lower mRNA levels in the NAM allele compared to B73, respectively (detected by DESeq2; Methods), while none of the B73 = NAM category were significantly different. Boxplots in **a**, **b**, **c** and **f** denote the range from the first to the third quartile, lines within boxes indicate the median and whiskers represent 1.5-fold of the interquartile range.

shared haplotype-specific MOA footprints at that site, compared to only 42.9% of the F1s with shared equally methylated alleles at that same site (Fig. 4c). The observed strong correlations between differential CG and CHG methylation and haplotype-specific MOA occupancy confirm an important role for DNA methylation in determining TF binding in maize.

### MOA bQTL explain a large portion of heritable variation

Regulatory variation is thought to underlie a significant proportion of phenotypic variation in maize[40]. To assess the relationship between bQTL and complex trait variation, we first quantified the enrichment of genotype-associated bQTL surrounding GWAS hits (lead SNP ± 100 bp) across two curated datasets of 41 and 279 traits[40,41]. Given that bQTLs showed a genome-wide distribution distinct from all SNPs (Supplementary Fig. 7), with bQTL located closer to genes, we generated a background dataset to match this distribution (similar allele frequency and distance to the nearest gene, 100 permutations; Methods) to avoid any bias caused by location in the genome. For both GWAS datasets tested, bQTL were approximately twofold (1.75-fold and 2.17-fold, respectively) enriched for co-localization with GWAS hits compared to the matched

bgSNPs (Supplementary Fig. 13). This enrichment remained stable as a function of distance to the nearest gene, indicating comparable efficacy of bQTL to mark functionally significant loci genome-wide (Fig. 5a). To explore the degree to which bQTL can more broadly capture the genetic variation underlying phenotypic diversity, we partitioned heritable trait variance for 143 traits in the NAM population (Methods and previous publications[2,17]). We modeled additive genetic variation for traits as a function of three genomic relatedness matrices. Variances estimated this way for several trait datasets simulated from matrices highly similar to our observed matrices accurately reflected the proportional contributions of each SNP set (Supplementary Fig. 14). Across a large majority of phenotypes in the NAM panel (103 of 143 or ~72%), bQTL associated with genotype alone (that is, excluding methylation, 78,398 bQTL) explained the majority of the total additive genetic variance captured by SNPs (Fig. 5b, Supplementary Fig. 15 and Supplementary Table 8). Consistent with previous findings that open chromatin and TF binding, found at a higher frequency close to genes, have a key role in trait variation[2,17], our matched bgSNPs (matched allele frequency and distance to the nearest gene compared to bQTL) often accounted for

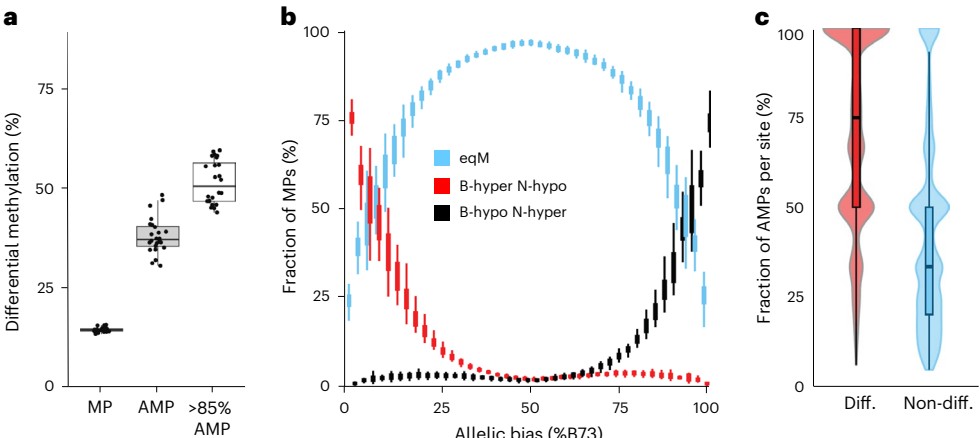

**Fig. 4 | bQTL are linked to variation in DNA methylation. a**, Genome-wide overlap of differentially methylated (CG and/or CHG) DNA regions with MPs, AMPs and AMPs with strong (≥85%) allelic bias, across the 24 F1s. **b**, Correlation of differentially CG-methylated DNA with allelic bias for MPs in the 24 F1 hybrids. MP methylation categories: equally methylated (eqM), B73 hypermethylated versus NAM hypomethylated (B-hyper N-hypo) or B73 hypomethylated versus NAM hypermethylated (B-hypo N-hyper). **c**, Correlation between MOA footprint

bias and differential methylation at loci that varied both in allele-specific footprint occupancy (≥1 F1s with AMP) and CG methylation (≥2 F1s with and without allele-specific methylation difference) between the 24 F1s. At each position, F1s were partitioned into those with either differential allelic CG methylation (red) or equal CG methylation (blue). Box and violin plots were drawn for the two categories, showing the distribution of percentages of F1s with haplotype-specific binding (AMPs). $n = 24$ F1s in **a**, **b** and **c**; boxplots were generated as in Fig. 3.

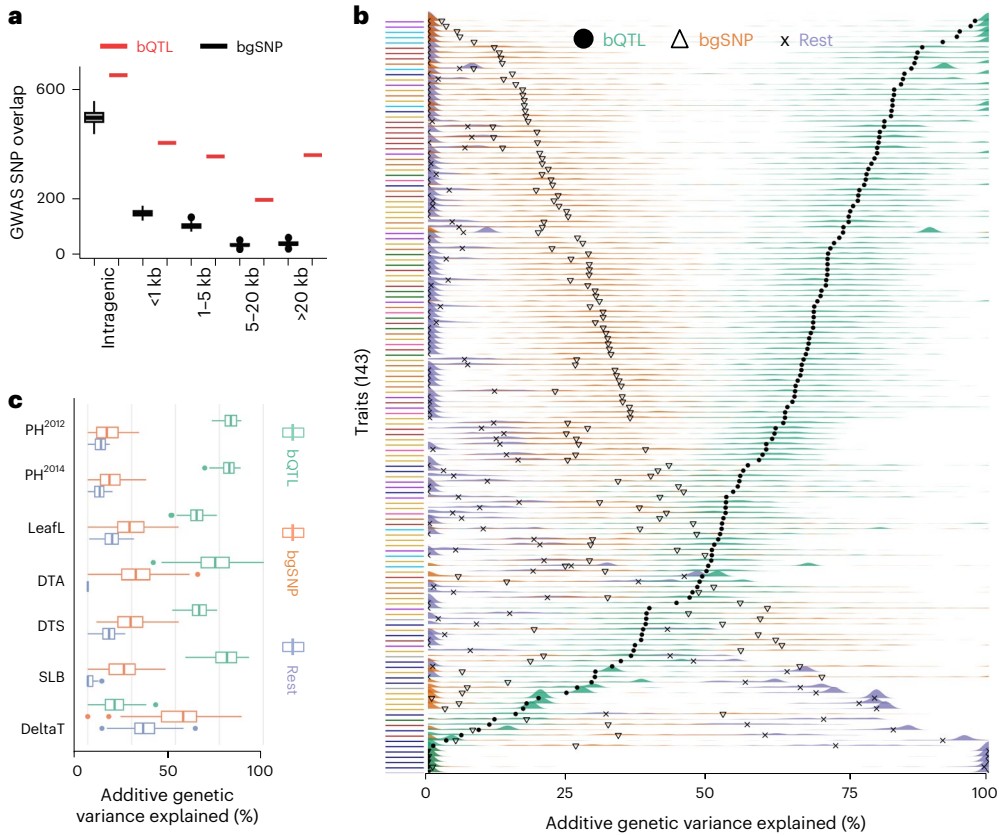

**Fig. 5 | A large fraction of heritability is explained by bQTL. a**, Association of ~42,000 curated GWAS hits[41] (±100 bp) with bQTL (only those associated with genotype alone), or $n = 100$ bootstraps of matched bgSNPs (same bgSNP sets used as for VCAP; Methods) at distances ranging from intragenic to >20 kb to the nearest gene. **b**, Estimated additive genetic variance organized by 143 traits. Colored ridges show the estimated additive genetic variance across 100 permutations for either bQTL, bgSNPs or remaining genome SNPs. Black symbols represent the mean estimated value across permutations. Traits are arranged by bQTL mean variance estimates and color-coded according to general trait

groupings: vitamin E metabolites, navy blue; metabolites, purple; stalk strength, light blue; flowering time, gold; plant architecture, red; disease, green; tassel architecture, pink; ear architecture, orange; miscellaneous, gray. **c**, A subset of traits (y axis) and their estimated percent additive genetic variance (x axis) shown as colored box plots instead of ridges. PH, plant height[51,52]; LeafL, leaf length[51]; DTA, days to anthesis[51,52]; DTS, days to silking[51]; SLB, southern leaf blight[53,54]; and delta-tocopherol concentration, vitamin E biosynthesis[43]; $n = 100$ permutations. Boxplots in **a** and **c** were generated as described in Fig. 3; data outside the whisker range are considered outliers.

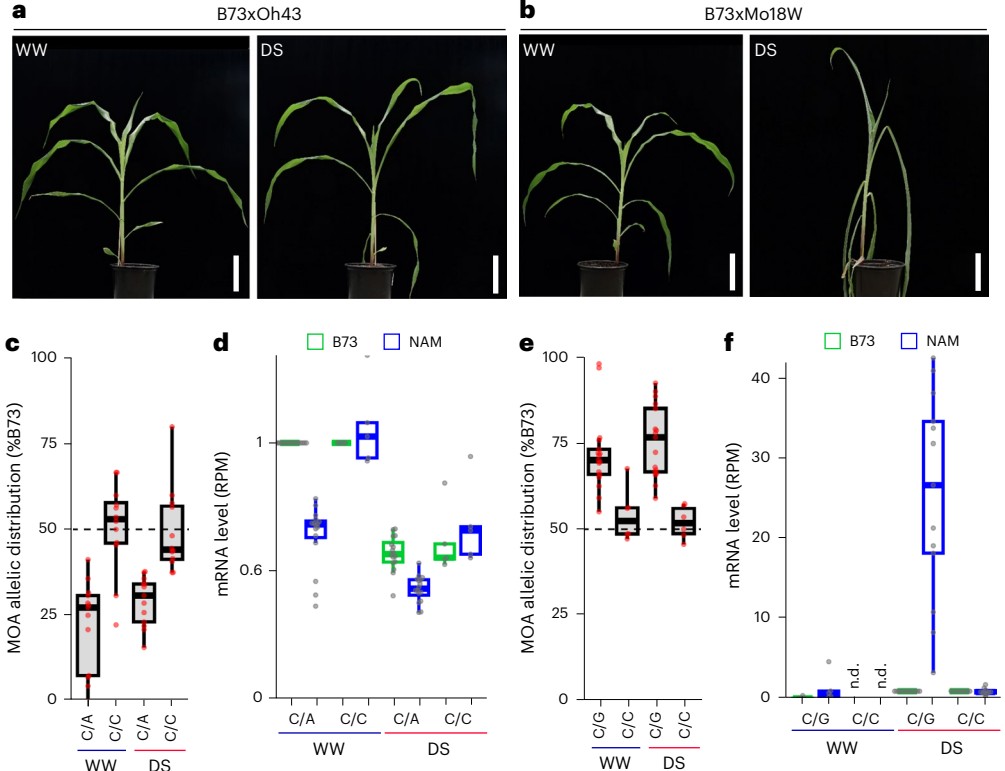

**Fig. 6 | Characterization of a drought-responsive cistrome. a,b,** Morphological phenotypes of the more tolerant B73 × Oh43 (**a**) and susceptible B73 × Mo18W (**b**) F1s grown under WW and DS conditions. Scale bars, 10 cm. **c**, Allelic distribution of MOA read coverage at bQTL (B73-chr1:198,205,029) in the *ZmSO* promoter. The six F1s sharing the B73 allele (C/C) were compared to 17 F1s carrying C/A alleles. **d**, Haplotype-specific mRNA counts, normalized to the B73 WW allele, grouped by the *ZmSO* bQTL alleles shown in **c**. Haplotype-specific read counts (reads per million (RPM), also adjusted for differences in SNP counts between F1s) relative to the B73 allele WW level to allow count comparisons between F1s (*n* = 21 F1s

with gene polymorphism to permit haplotype-specific analysis, 5 C/C and 16 C/A alleles). **e**, Allelic distribution of MOA reads at a bQTL (B73-chr2:28,118,442) in the *ZmTIP3d* promoter. The 12 F1s sharing the B73 allele (C/C) were compared to 13 F1s carrying C/G alleles. **f**, Haplotype-specific mRNA abundance grouped by the *ZmTIP3d* bQTL alleles shown in **e**. RPM values per haplotype are normalized to the B73 DS mRNA level (*n* = 23 F1s that permitted haplotype-specific MOA/RNA analysis: DS, 10 C/C, 13 C/G; WW: 5 C/C, 5 C/G). n.d., not detected. Boxplots in **c**, **d**, **e** and **f** were generated as described in Fig. 3.

more additive genetic variation than SNPs from the rest of the genome (that is non-bQTL, non-matched bgSNPs; 121 out of 143 traits), but bQTL also outperformed bgSNPs for most traits (81.1%, 116 traits; Fig. 5b). The inclusion of bQTL with additional significantly associated differential methylation (105,398 bQTL) slightly decreased the variation explained (Supplementary Fig. 16). This is consistent with theoretical arguments that epigenetic variation, which is highly labile on an evolutionary timescale, cannot explain much heritability for phenotypes[42]. Traits for which bQTL explained the largest portion of genetic variance included plant height, leaf size or shape and disease resistance, whereas almost all traits related to, for example, vitamin E production were best explained by the bQTL-matched bgSNPs or the remaining SNPs from the rest of the genome (Fig. 5c), probably because of the oligogenic nature of the vitamin E traits and that bQTL identified in leaf tissue may not be representative of regulatory patterns in genes specifically expressed in kernels[43].

## Characterization of a drought-responsive cistrome

To evaluate differences in *cis*-element regulation induced by changes in environmental conditions, we compared the morphological and molecular response of our F1 population under well-watered (WW) and drought-stress (DS) conditions. We observed diverse drought responses, with reductions of relative leaf water content of 3–30% and remaining soil water contents of 6.3–25.6%, depending on the F1 (Fig. 6a,b, Supplementary Fig. 17 and Extended Data Fig. 4). Haplotype-specific MOA-seq and RNA-seq of WW and DS samples for all 25 F1s revealed on average 287,844 MPs and 56,863 AMPs under

DS, slightly less than for WW conditions (Supplementary Table 5), and a similar correlation with allele-specific transcript abundance (Supplementary Fig. 10). MOA peaks showing significant (*P* < 0.05) drought-induced increases or decreases in occupancy varied substantially among F1s, ranging from ~9,000 to 40,000 and 16,000 to 90,000, respectively (Supplementary Table 9). Local association mapping identified 124,504 DS-bQTL for SNPs and 23,554 for small INDELs under drought conditions (Supplementary Table 10), for a combined total of 206,368 unique SNP bQTL in DS and/or WW. To identify candidate drought-response loci, we selected bQTL with drought-responsive occupancy near genes (5 kb upstream or 1 kb downstream) that displayed both haplotype-specific and drought-responsive transcript accumulation, resulting in 1,025 (655 genes) and 2,604 (1,548 genes) bQTL with increased and decreased occupancy, respectively. Further integration with drought-response GWAS and *cis*-expression QTL hits[44–46] resulted in high-confidence candidates (Supplementary Table 11). Notably, the candidate list included known drought-tolerance-related genes, such as *ZmNAC111* (Zm00001eb405590). Haplotype-specific MOA footprinting identified multiple DS-bQTL upstream and downstream of *ZmNAC111*, including adjacent to the causative 84 bp MITE transposon insertion site, which reduces both *ZmNAC111* expression and drought tolerance in maize seedlings, probably through RNA-directed DNA methylation[47] (Supplementary Fig. 18). Another interesting DS-bQTL was located within the previously discovered 119 bp proximal promoter fragment required for the drought-response of SULFITE OXI-DASE1 (*ZmSO*, Zm00001eb036560), a gene linked to the ABA response and drought tolerance of maize seedlings[48]. Although none of our

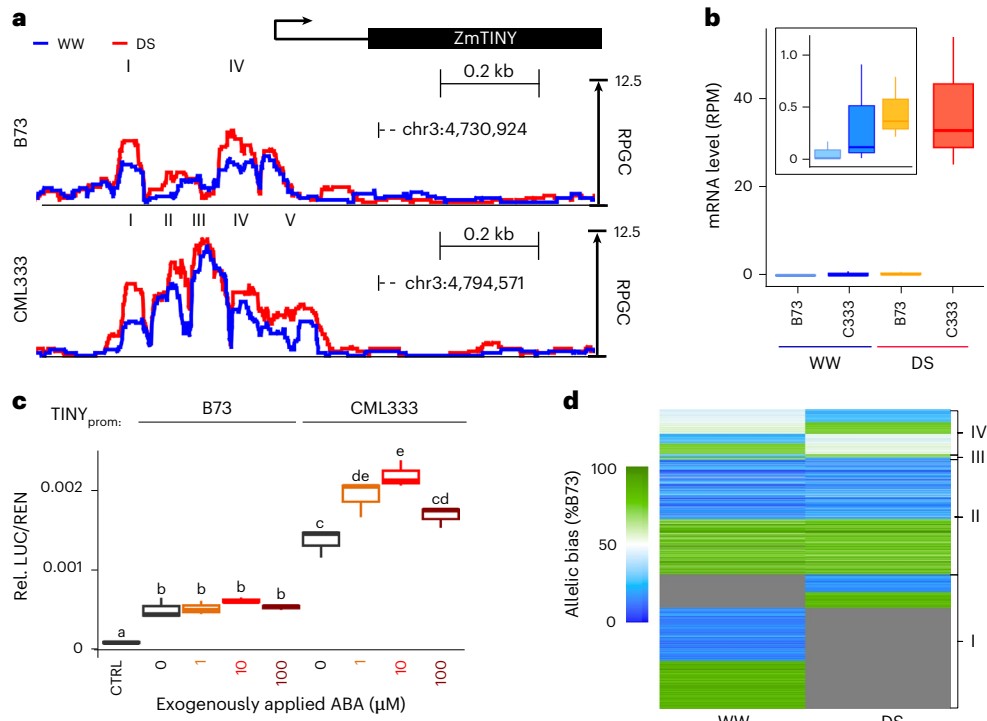

**Fig. 7 | MOA-seq detects drought-responsive *cis*-regulatory loci. a,** Average (three biological replicates), normalized MOA coverage for B73 × CML333 near *ZmTINY* under WW and DS conditions. Roman numerals highlight respective DS-peak regions identified (*P* < 0.05, MACS3) in each line. **b,** Haplotype-specific mRNA counts of *ZmTINY* for B73 × CML333 (C333) under WW and DS conditions (three biological replicates). The zoomed-in square highlights (left to right) WW-B73, WW-CML333 and DS-B73. **c,** Relative luciferase levels in maize leaf protoplasts harboring either ~0.5 kb of the B73 or CML333 proximal promoter allele upstream of the *ZmTINY* TSS, with and without 1–100 µM ABA treatment. Letters represent significant differences (*P* < 0.05, ANOVA with Tukey test; Source data). **d,** Heatmap of MOA allelic bias at AMP loci under WW and DS conditions. AMPs (under WW, DS or both) in drought-responsive MOA peaks

(*P* < 0.05; Methods) are displayed for B73 × Oh43. Color scale ranging from green (100% bias towards B73) to blue (100% bias towards Oh43); gray represents MOA signal below the detection limit (MACS3; Methods). Clusters represent (I) allele-specific occupancy in one condition and below detection limit in the other, (II) allele-specific occupancy with consistent bias under both conditions, (III) allele-specific occupancy with bias in the opposite direction under the two conditions and (IV) occupancy with a significant allele-specific bias under only one condition. Only sites with significant allele-specific bias (binomial testing, FDR, 1%; for details see Methods) in at least one condition were considered. To avoid additional statistical cut-off effects for the second condition, 60% or more occupancy bias towards one allele was considered allele-specific. Boxplots in **b** and **d** were generated as described in Fig. 3.

haplotypes contained the putative Myb-binding site (CAGTTG) previously linked to drought-response in the 119 bp *ZmSO* promoter[48], we nonetheless found a strong correlation between increased MOA occupancy for the C allele at the bQTL and elevated *ZmSO*[B73] transcript levels, both under WW and DS conditions (Fig. 6c,d). We also observed a strong correlation between MOA occupancy and drought-induced transcript levels at DS-bQTL in the proximal promoter of the maize homolog of aquaporin BETA-TONOPLAST INTRINSIC PROTEIN 3 (*ZmTIP3d*, Zm00001eb076690; Fig. 6e,f), which has been linked to drought-response in various plants[49].

To further test the correlation of DS-bQTL and drought-responsive promoter activity, we analyzed the maize homolog of the *Arabidopsis* drought-inducible AP2/ERF TF AtTINY independently in a transient expression assay. Over-expression of *AtTINY* increases drought tolerance at the cost of severely stunted growth, a limitation often observed with drought-related TFs[50]. The maize homolog of *ZmTINY* (Zm00001eb120590) is a candidate gene for drought response and leaf size variation[45]. We found DS-bQTL in multiple MOA footprints surrounding *ZmTINY*, which showed significantly higher occupancy in, for example, CML333 and Oh43 compared to B73 under drought (ranging from 1.4-fold to 5.5-fold higher; Fig. 7a and Extended Data Fig. 5). Similarly, higher MOA occupancy under DS for CML333 and Oh43 compared to B73 was also observed downstream of *ZmTINY* (Extended Data Fig. 5). These variations in MOA footprints were correlated with allele-specific transcript levels of *ZmTINY*. In F1s under DS conditions, mRNA transcripts of the CML333 and Oh43 alleles were 84-fold and 18-fold more

abundant than B73 transcripts, respectively (Fig. 7b and Extended Data Fig. 5). MOA signals in the B73 and CML333 upstream promoter showed the highest correlation to *ZmTINY* mRNA levels (Extended Data Fig. 5). We tested these sequences in a dual-luciferase expression assay with and without ABA treatment to simulate DS. Both promoter fragments exhibited significantly higher LUC/REN ratios than the vector control. Consistent with trends observed for MOA and mRNA levels (Fig. 7b,c), prom::*ZmTINY*[CML333] showed a higher LUC/REN ratio than prom::*ZmTINY*[B73] under WW conditions, and exogenous application of 1 µM and 10 µM ABA further increased the LUC/REN ratio significantly for protoplasts harboring the prom::*ZmTINY*[CML333] but not prom::*ZmTINY*[B73] fragment by 41.4% and 60.3%, respectively. Together, the results support previous findings of *ZmTINY* as a drought candidate gene and indicate that bQTL can identify *cis*-regulatory regions that act condition-dependently. That said, the drought-responsive regulation of *ZmTINY* may include additional regulatory sequences, such as the drought-responsive loci downstream.

Differences in MOA-seq coverage between WW and DS conditions at allele-specific sites could be caused by changes in occupancy level, the direction of allelic bias or a combination thereof. To better understand which scenario is more common, we clustered drought-responsive AMPs in the B73 × Oh43 F1 (11,970 AMPs located in drought-responsive footprints). The results showed that 83% of drought-responsive AMPs changed MOA occupancy between WW and DS conditions, either from no detectable MOA signal to haplotype-specific binding (~45%, group I), or in the amount of MOA coverage between WW and DS conditions while maintaining their allelic bias (~38%, group II) (Fig. 7d). By contrast,

only about 17% of AMPs showed bias changes, either from no significant bias in one condition to a significant bias in the other (~15%, group IV) or changing the direction of the allelic bias (~2%, group III) (Fig. 7d). Although groups I and IV are somewhat dependent on statistical cut-offs (peak calling and thus AMP definition and/or calling allelic bias), groups II and III show an allele-specific bias under both conditions. Focusing on groups II and III, it becomes evident that changes in allelic bias are ~20-fold less frequent compared to the constant binding bias accompanied by overall changes in MOA signal. Similar clusters between WW and DS conditions were observed for AMPs in all 25 F1s (Supplementary Fig. 19). We therefore propose that the majority of DS-induced TF occupancy dynamics at sites of functional genetic variation results from condition-specific TF abundance changes rather than changes in allelic bias between WW and DS conditions.

## Discussion

The gene regulatory landscape involves primary sequence, chromatin accessibility and DNA and protein modifications[10]. Although our ability to assemble complex genomes has made great progress, decoding gene regulation, population-wide, high-resolution maps of the regulatory loci and efficient pinpointing of functional variation remain elusive in plants[5].

We present a robust, high-throughput method for identifying functional variants, genetic, epigenetic, or both, linked to trait variation in plants. By integrating haplotype-specific TF footprints and transcript abundance, F1 hybrids and local association mapping at putative *cis*-element loci, we defined a pan-cistrome of the maize leaf under WW and DS conditions. Use of concatenated dual-reference genomes and F1 hybrid analysis resolved issues of reference bias, *trans*-effects and technical variation that commonly compromise haplotype-specific quantitation. Although MOA-seq footprints with their high-resolution (~100 bp) and comprehensive cistrome-wide analysis are well suited for this method, similar results may be obtained by, for example, allele-specific ATAC-seq (with putatively a lower fraction of the genome covered owing to the large size of the Tn5 dimer[11,13]) or for single TFs using allele-specific ChIP–seq, as we demonstrated previously in one F1 hybrid[17].

Our analysis demonstrates a high level of variation in *cis*-regulatory networks among 25 diverse maize genotypes and provides a high-resolution map of regulatory elements underpinning the function of over 200,000 putative *cis*-element loci in the maize leaf. We note that the high genetic diversity between maize inbred lines allowed us to detect variants in 25 F1 lines. For species with lower diversity, more F1s or the inclusion of more distant interspecies hybrids might be necessary. Finally, we highlight the relevance of genotype-associated bQTL for understanding phenotypic diversity in maize, demonstrating that haplotype-specific MOA-seq in leaves allowed us to capture the majority of additive genetic variation for most tested phenotypes.

## Online content

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

[1]Heinrich Heine University Düsseldorf, Faculty of Mathematics and Natural Sciences, Institute for Molecular Physiology, Düsseldorf, Germany. [2]Independent Research Groups, Max Planck Institute for Plant Breeding Research, Cologne, Germany. [3]DIADE, University of Montpellier, CIRAD, IRD, Montpellier, France. [4]Department of Ecology, Evolution, and Organismal Biology, Iowa State University, Ames, IA, USA. [5]Cluster of Excellence on Plant Sciences (CEPLAS), Heinrich Heine University, Düsseldorf, Germany. [6]Heinrich Heine University Düsseldorf, Faculty of Mathematics and Natural Sciences, Institute for Quantitative Genetics and Genomics of Plants, Düsseldorf, Germany. [7]Department of Biological Science, Florida State University, Tallahassee, FL, USA. [8]Perelyn, Munich, Germany. [9]School of Integrative Plant Sciences, Plant Breeding and Genetics Section, Cornell University, Ithaca, NY, USA. [10]Department of Plant Sciences, University of California, Davis, CA, USA. [11]Swammerdam Institute for Life Sciences, University of Amsterdam, Amsterdam, Netherlands. [12]Department of Agronomy and Horticulture, University of Nebraska–Lincoln, Lincoln, NE, USA. [13]Maize Research Institute, Sichuan Agricultural University, Wenjiang, China. [14]Institute of Bio- and Geosciences (IBG-2: Plant Sciences), Forschungszentrum Jülich, Jülich, Germany. [15]Institute of Bio- and Geosciences (IBG-4: Bioinformatics), Bioeconomy Science Center (BioSC), Forschungszentrum Jülich, Jülich, Germany. [16]Julius Kühn Institute, Federal Research Centre for Cultivated Plants, Institute for Breeding Research on Agricultural Crops, Groß Lüsewitz, Germany. [17]Institute for Transformative Biomolecules, ITbM, Nagoya University, Nagoya, Japan. [18]Department of Evolution and Ecology, Center for Population Biology and Genome Center, University of California, Davis, CA, USA. ✉e-mail: hartwit@hhu.de

## Methods

### Statistics and reproducibility

Experiments were conducted with three biological replicates unless otherwise stated. Pan-cistrome analysis was performed on 26 genomes (25 paternal and one maternal genome). Data collection and analysis were not performed blind to the conditions of the experiments. For randomization of plant positions of drought-treated plants, see the plant materials section. No data were excluded from the analyses. Data distribution was assumed to be normal, but this was not formally tested.

### Plant materials

The GRIN National Agricultural Library supplied B73, Mo17, A619, W22, A188 and US-NAM seeds. Seeds were pre-germinated for 48 h at 28–30 °C. Each pot contained soil equalized by volume and four seedlings (Einheitserde VMV800/D373 soil). Plants were grown in a greenhouse using a randomized block design, under long-day conditions (16 h day, 8 h night, 28–30 °C) for approximately 26 days until 75% of the plants per genotype showed the formation of the leaf four auricle. Plants were then randomized, and 12 plants (three pots) per treatment and replicate were grown with or without periodic watering through a bottom drench system for 86 h. Plants were then harvested, and the leaf blades of the oldest leaf without a yet-formed auricle were immediately frozen in liquid nitrogen. Additionally, the relative water content and soil water content were measured (Supplementary Note 2). No statistical methods were used to pre-determine sample sizes, but our sample sizes are similar to those reported in previous publications[17].

### MOA-seq and RNA-seq sample preparation and sequencing

MOA-seq and RNA-seq sample and library preparation were performed as previously described[11,17]; for details, see Supplementary Note 3.

### MOA-seq data analysis

Reads were filtered using SeqPurge[55] (v.2022-07-15) with parameters '-min_len 20 -qcut 0'. Owing to the short fragment length in MOA, read pairs almost completely overlapped. MOA-seq paired-end reads were merged into single-end reads, including base quality score correction, using NGmerge[56] (v.0.3) with parameters '-p 0.2 -m 15 -d -e 30 -z -v'. Diploid genomes were created by concatenating the B73 v5 genome with the respective paternal genome (NAM v1/2 genomes[18], Mo17 CAU v1, W22 v2 (ref. 20), A188 v1 (ref. 19) and A619, with accession ID added as a prefix to the chromosome name; Supplementary Note 4). Reads were mapped to the diploid genome (or the separate genomes for inbred data) using STAR[57] (v.2.7.7a). STAR was designed to map RNA; therefore, we used the flag –alignIntronMax 1 for DNA (no introns allowed) as well as parameters '–outSAMmultNmax 2, –winAnchorMultimapNmax 100' and '-outBAMsortingBinsN 5'. We generated two datasets: one in which reads were only allowed to map once in the diploid genome (mapping quality 255, used to generate MPs and AMPs data) and one in which reads mapped exactly twice, with double mapping reads being randomly assigned to one of the two positions (used for visualization and overall peak coverage data). Format conversion and calculation of the average mapped fragment length (AMFL) were done using SAMtools[58] (v.1.9). The effective genome size was calculated using unique-kmers.py (https://github.com/dib-lab/khmer, commit fb65d21), with AFML and respective genome fasta as inputs. The deeptools[59] (v.3.5.0) function bamCoverage was used to generate normalized (reads per genome coverage (RPGC)) bedgraph files of full-length read data.

Fragment-center tracks were generated as previously described[60]: bam files were converted to bed format using bamToBed of Bedtools (v.2.29.0)[61], and each mapped read was shortened to 20 bp centered around the middle of the read using awk; reads with an uneven number of bases were extended 10 bp to each site from the middle of the read. One of the two middle bases was chosen at random for reads with even number of bases, and reads were extended 10 bp to each site. The function genomeCoverageBed of Bedtools was then used to convert the

bed files to bedgraph, scaled by the quotient of the effective genome size and the number of uniquely mapped reads (similar to RPGC of deeptools bamCoverage). BigWig files for visualization were generated using bedGraphToBigWig (v.4)[62].

### MP and AMP identification

To enable translation between coordinates of the B73 genome and the paternal genomes, hal files were generated using the cactus function of progressive cactus[63] (v.1.0.0, 2020-04-19) with standard parameters. SNPs between B73 and the paternal lines were determined with the halSnps function of cactus, using parameters 'unique' and 'noDupes'. From the resulting SNP lists, we selected all SNPs that carried either the B73 or one other base in all analyzed lines (biallelic SNPs, 117,898,189). Of the remaining SNPs, only those occurring in at least two of the 25 parental lines were retained (60,432,443; minor allele frequency, 0.08). For allele-specific analysis, B73 coordinates of the filtered, biallelic SNPs were translated back to paternal coordinates using halLiftover (hal-release-V2.1), and all SNPs with ambiguous corresponding positions in one of the two parental genomes were removed (de-duplicated biallelic SNPs in at least two lines, 58,823,746). At each of these SNP positions, we counted RPGC values for both alleles using bedtools map (bedtools v.2.29.0) and calculated the read numbers corresponding to the RPGC numbers for further calculation (for example, binomial testing was performed on read numbers, not normalized values). Binding frequencies at SNP positions were determined as RPGC-B73 / (RPGC-B73 + RPGC-Pat). We defined MPs as SNPs that were located within MOA peaks and had more than seven RPGC (approximately >25 reads) for at least one allele and at least one read in the corresponding allele. Allele-specific binding at MPs (significant deviation from the expected 0.5 binding frequency) was determined by binomial testing in R (v.4.1.1). SNP positions with an FDR-corrected $P$ value of <0.01 were considered AMPs. Additionally, we determined the allelic ratio of WGS control reads (Supplementary Note 5) in a 65 bp window at all MPs, and excluded all AMPs with a WGS ratio above or below the upper and lower fifth percentile value of all MPs, respectively.

### Peak calling

For peak calling, MOA bam files were used with MACS3 (v.3.0.1, https://github.com/macs3-project/MACS) using the following parameters: -s and –min-length 'AMFL', –max-gap '2x AMFL', –nomodel, –extsize 'AMFL', –keep-dup all, -g 'effective genome size', where AMFL represents average MOA fragment length, calculated with SAMtools stats using default parameters.

### Treatment-specific peak calling

MOA-alignment bam files were converted to bed format using bedtools bamToBed (v.2.29.0). The genomeCoverage function of bedtools was used to convert pooled replicated bed files to bedgraph with the reads per million scaling factor. The reads-per-million-normalized coverage difference between treatments was calculated using the intersect and subtract functions of bedtools. The resulting differences in coverage counts for WW and DS treatments were used to create an unbinned (1 bp bin) bed file to produce a bigwig coverage track, which was used as input for MACS3 (v.3.0.1) peak calling, using parameters: –min-length 30, –max-gap 60, –nomodel, –extsize 1, –keep-dup all, -g 'effective genome size', -q 0.01. Significant differences between WW and DS peaks were determined by a two-sided Welch $t$-test using the individual bio replicate coverages, and peaks with $P < 0.05$ were retained.

### Transient luciferase assay

Protoplasts were isolated and transformed by electroporation as previously described[64] (Supplementary Note 6) using 10 μg of a plasmid encoding firefly luciferase downstream of the respective B73 or CML333 prom::TINY alleles (B73 fragment +570 bp upstream of ATG or CML333 + 451 bp upstream of ATP; primers in Supplementary Table 12),

along with 5 µg of a plasmid containing 35S-renilla luciferase. For ABA treatments, a 20 mM stock solution of (+)-Cis, Trans-Abscisic Acid (Duchefa Biochemie, cat. no. A0941) in ethanol was prepared. Round-bottom 2 ml microcentrifuge tubes were pre-loaded with 50 µl of ABA solution in protoplast buffer[64] (Supplementary Note 6), achieving the required ABA concentration upon the addition of 950 µl of electroporated protoplasts. After transformation, the protoplasts were incubated for 18–22 h in the dark for recovery. The cells were sedimented for 2 min at 260$g$ at room temperature and resuspended in 80 µl of 1× Passive Lysis Buffer (Promega, cat. no. E1941). Cells were disrupted by vortexing for 10 min, and cell lysates were cleared by centrifugation for 10 min at 12,000$g$ at 4 °C.

The dual-luciferase assay was performed as previously described[65] (Supplementary Note 7). All experiments were conducted with three biological and three technical replications. Values were calculated by dividing the activity of firefly by that of renilla luciferase.

## DNA methylation analysis

Parental DNA methylation data of the NAM lines[18] were obtained from iPlant (/iplant/home/maizegdb/maizegdb/NAM_PROJECT_JBROWSE_AND_ANALYSES). Methylation data for non-NAM lines[19,66] were obtained as SRA archives (Bioprojects PRJNA657677 and PRJNA635654) and processed as previously described[17] (Supplementary Note 8). B73 × Mo17 hybrid methylation data were previously published[17] and showed strong correlation with parental methylation at B73 × Mo17 AMPs (Extended Data Fig. 2a–d). Context-specific methylation around AMPs and MPs was determined separately for the B73 and paternal alleles in the three sequence contexts (CG, CHG or CH) as the averaged methylation levels within a window of ±20 bp around the position as previously described[17]. Significant differences in DNA methylation were determined following a previous publication[39] (one allele <10% methylated and the other >70%). Sites for testing the consistency of DNA methylation or haplotype-specific binding relations among the F1 hybrids were selected based on having at least two F1 lines differentially methylated, at least two F1 lines equally methylated and at least one F1 line AMP at the given site. In this analysis (Fig. 4c), a more stringent definition of equal methylation (as opposed to not being differentially methylated) was used: equal methylation was defined as both alleles <10% or both >70% methylated.

## Local association mapping to map bQTL

The binding ratio of the MOA peaks, as well as methylation ratio information for mCG, mCHG and mCHH, were collected separately for all hybrids for the WW and DS conditions. The binding frequency at loci with no reads was set to 'NA'. Genotyping information ($GT_i$) at sequence variants and the methylation ratio information were used to conduct local association studies using five different linear models for each MP. All MPs with the respective haplotype-specific MOA coverage (binding frequency) and average surrounding (±20 bp) methylation ratios were considered:

$$MP = mCHH$$

$$MP = mCG$$

$$MP = mCHG$$

$$MP = GT_i$$

$$MP = GT_i + mCG + mCHG + mCHH$$

The analyses were performed in Julia (v.1.8.1) and R (v.4.4.1). Associated MPs at a FDR of 5% were selected (R v.4.1.2). bQTL located within 65 bases were combined into linkage groups (the lowest $P$ value determined the lead bQTL).

## Analysis of MPs at INDELs

A list of candidate INDELs was generated through pairwise whole-genome alignment of each of the 25 inbred parental genomes to the common mother's, B73, genome. Alignments were created using Anchorwave's (v.1.2.2) proali function[67], with anchor regions determined with minimap2 (ref. 68) (v.2.27-r1193). Variants were called from the alignment using wgatools' (v.0.1.0[69]) call function using the -s and -l 1 parameters to call variants of any size. For later ease of handling, variants were extracted from the created variant call format file and written into BED (browser extensible data) file format. Liftover of coordinates from B73 to each of the other 25 parental genomes was facilitated with CrossMap[70] (v.0.7.0), using chain files created with wgatools maf2chain[69]. The following steps were performed using custom code as well as the bedtools suite[61] (v.2.30). From the list of variants, a set of INDELs between 2 and 50 bp in length and biallelic in the population was extracted. In addition, this set was filtered so that the non-B73 allele occurs at least twice in the population. Synteny analysis of INDELs by whole-genome alignment across genomes is challenging; hence, we tried to minimize ambiguously mapped INDELs with additional filters[71].

RPGC values for B73 and NAM INDEL regions were determined using bedtools intersect (-wao parameter; bedtools v.2.29.0) on bedgraphs containing the normalized read counts determined as described above and the bed file containing the INDEL positions. The -wao function returns, for each bed entry, the overlapping bedgraph entries, including the length of the overlap, even if the entry count is zero. From this result, custom awk commands were used to calculate the average RPGC (sum over all bases/length of region, including bases with zero count) for the B73 and parental allele. For each deletion allele, average RPGC values were calculated over a window of 3 bp before and 3 bp after the deletion. For each insertion allele, average RPGC values were calculated over the whole insertion itself. After this step, counts were treated in the same way as SNP counts. The same methylation data and analysis steps were used for INDELs as for SNPs (see above), with methylation being calculated in a window from 20 bp upstream of the insertion or deletion start to 20 bp downstream of the insertion or deletion end coordinate. Methylation and count data were then used to perform bQTL analysis in the same way as described for SNPs.

## RNA-seq analysis

RNA-seq data were mapped to the concatenated diploid genomes using STAR (v.2.7.7a), with options –outSAMmultNmax 1, –outFilterMultimapNmax 1, –winAnchorMultimapNmax 100, –twopassMode Basic, –outFilterIntronMotifs RemoveNoncanonical, –outFilterType BySJout, –quantMode GeneCounts, using a concatenated gff3 file containing gene models from both parents. To determine allele-specific transcript abundance, for each line, B73 and corresponding paternal positions for all SNPs determined by halSnps were generated by halLiftover. Of the resulting position pairs, ambiguous ones (mapping to more than one position in one of the genomes) were removed. Each SNP was then assigned the B73 genes it overlaps. The respective NAM gene info was added using a Pan-gene file (downloaded from MaizGDB), retaining strand information in both cases. Mapped read information was converted into read bed files using bamToBed, and each SNP was assigned all reads overlapping with it in B73 and at the parental genome coordinates (strand-specific, separately for the three replicates). Only SNPs carrying reads in both alleles were retained to ensure that the SNP was truly located within the gene in both alleles. Afterwards, reads for each gene were counted per replicate (reads that had two or more SNPs were counted only once) and allele (Supplementary Table 13a,b). For A188, for which no Pan-gene entries were available, SNPs were also mapped onto the A188 gff3, and gene pairs were generated based on shared SNP positions. In this way, B73 reads and paternal reads could be compared for differential transcript abundance analysis in DEseq2 (ref. 72) in R (v.4.1.1). Genes with an FDR-corrected $P$ value of <0.05 were considered to have ASE in their transcript abundance.

## Variance component analysis pipeline

To run the variance component analysis pipeline (VCAP), we required three datasets: genome-wide markers across the NAM population recombinant inbred lines (RILs); trait values across NAM RILs for each trait analyzed; and coordinates for MOA peaks or bQTL SNPs across founder lines to partition each component. For the genome-wide markers, we used publicly available resequencing SNPs from the NAM founders[18] that had been projected onto the NAM RILs (/iplant/home/shared/NAM/Misc/NAM-SV-projected-V8). Trait data collected from the NAM RILs ($n$ = 143) were curated from previous publications[73] (Supplementary Table 14). We used two sampling schemes to create our MOA partitions. First, only the bQTL SNPs with significant association to the genotype, not methylation, were used to represent MOA. Second, bQTL associated with genotype and methylation at the same time were included. Any SNP outside of the bQTL SNPs created the non-MOA pool from which the bgSNPs were drawn. Given the non-random distribution of bQTLs throughout the genome, we also included a matched background component: each bQTL SNP was matched to a random non-bQTL SNP by allele frequency (number of lines containing the alt allele / total lines without missing data at that position, 0.1 bin size) and distance from the nearest gene (TSS or transcription termination site as calculated by bedtools[61] closest -d).

This matched set of MOA and bgSNPs, equal in number to SNPs with similar genomic contexts, was used for a single VCAP run. Kinship matrices were created for the bQTL SNPs, bgSNPs and the rest of the genome (remaining non-bQTL and non-bgSNPs) using Tassel (v.5)[74]. To calculate the heritabilities of all 143 traits, the set of three kinship matrices and traits was run through a REML model using LDAK (v.5.2)[75]. We sampled 100 times, creating 100 permutations of kinship matrix sets. Thus, the permutations gave us a range of heritability estimates that could result from these particular components, traits and the population (Fig. 5b). The same bQTL SNPs were used in every permutation, whereas the bgSNPs differed across permutations.

To evaluate the reliability of our heritability estimation method, we simulated traits with defined contributions from specific sets of kinship matrices and compared estimates of the heritabilities generated by the above VCAP protocol. We used one of our previously generated kinship matrix sets (one SNP per peak sampling) to simulate traits assigned certain heritabilities for each component (four sets of heritabilities, ten traits per set). We simulated traits as the sum of four normally distributed random vectors, each with zero mean and covariance equal to one of the three kinship matrices or the identity matrix (for residual variation) multiplied by a specific heritability value. The simulated traits and kinship matrices were used in the REML modeling step to estimate heritabilities. Estimated heritabilities were then compared to known heritabilities. All scripts written for the analyses in the study were deposited at https://github.com/Snodgras/MOA_Analysis.

## MOA bQTL and eQTL linkage analysis

Linkage disequilibrium was calculated between the binding QTL reported in this study and a set of 10,618 cis-eQTL identified based on expression data of the roots of 340 maize genotypes[38]. Genomic coordinates of the 78,398 binding QTL on the B73_RefGen_V5 maize genome were converted to B73_RefGen_V4 positions using CrossMap (v.0.6.4) as implemented in EnsemblPlants[70,76]. A total of 99.4% of bQTL positions could be successfully converted to B73_RefGen_V4 positions, and of these, 38,291 were present in a set of 12,191,984 genetic markers segregating in the population of 340 maize lines used to conduct eQTL analysis with a minor allele frequency of ≥0.05 and less than 2% of genotypes exhibiting heterozygous genotype calls. Linkage disequilibrium was calculated between bQTL markers and cis-eQTL markers in all cases in which a cis-eQTL and a bQTL were located within 10 kbp of each other, using genotype calls from the 340 maize varieties[38,77]. To assemble a control set of genetic markers with the same properties as the bQTL, bQTL that were successfully converted to B73_RefGen_v4 and matched

to Sun et al.[38] markers were divided into ten bins based on their distance from the closest annotated TSS (0–1 kbp, 1–2 kbp and so on), plus two additional categories for intragenic SNPs and SNPs > 10 kbp from the nearest annotated gene. A random subset of two million B73_RefGen_v5 SNPs used to detect bQTL were also converted to B73_RefGen_v4 and matched to segregating markers from Sun et al.[38], as described above. These markers were subsampled to create a second set of 38,291 control markers with representation in each of the 12 bins equal to the levels observed for the real bQTL.

## Further data processing

To obtain the high-confidence list of drought-responsive MOA regions, all MOA bQTL (unclumped WW or DS) were filtered for overlap with AMPs located in regions with significantly ($P$ < 0.05) increased or reduced MOA occupancy between DS and WW conditions in at least two F1s (overlap within 65 bp). A total of 3,198 and 11,060 drought-induced and repressed loci, respectively, were retained.

Comparisons and calculations of lists were either performed in bedtools intersect or with custom awk and bash scripts. Hypergeometric tests for over-representation or under-representation, ANOVA and data visualization were performed in R. Pearson correlation coefficients of bigwig file format MOA-seq data were calculated and visualized using the multiBigwigSummary and plotCorrelation functions of deepTools[59] with a window size of 1,000 bases.

## Reporting summary

Further information on research design is available in the Nature Portfolio Reporting Summary linked to this article.

## Data availability

All raw MOA-seq and RNA-seq data discussed in this publication have been deposited at NCBI SRA under accession number PRJNA1101486. MOA coverage tracks and peak files have been deposited to the Gene Expression Omnibus under accession number GSE294039. Coordinates in processed data files are based on the concatenated genomes (chromosome names: LineID-chr), which, for convenience, were deposited at Zenodo (https://doi.org/10.5281/zenodo.15177272 (ref. 78)). Coverage and binding frequency data for all bQTL is accessible at maizegdb (https://jbrowse.maizegdb.org), at a custom browser (https://www.plabipd.de/ceplas/?config=maize_hartwig_config.json) and at Zenodo (https://doi.org/10.5281/zenodo.15177272 (ref. 78)). Previously published datasets used in this study include /iplant/home/maizegdb/maizegdb/NAM_PROJECT_JBROWSE_AND_ANALYSES ref. 18, SRA accession numbers PRJNA961163 (ref. 14), PRJNA657677 (ref. 66), PRJNA635654 (ref. 19), PRJNA311133 (ref. 20), PRJEB31061 (ref. 18), PRJNA10769 (ref. 79), PRJNA540700, PRJNA565870, PRJNA531553, PRJNA399729, PRJNA389800 (ref. 80) and SRP011907 (ref. 1). Source data are provided with this paper.

## Code availability

Custom scripts have been deposited to Github repositories (https://github.com/Snodgras/MOA_Analysis, https://github.com/corn2code/bQTL, https://github.com/jengelhorn/AS-MOA, https://github.com/jengelhorn/AS-RNAseq, https://github.com/Ako31415/FIND-CIS-analysis) and Zenodo (https://doi.org/10.5281/zenodo.15098013 (ref. 81), https://doi.org/10.5281/zenodo.15225769 (ref. 82), https://doi.org/10.5281/zenodo.15097609 (ref. 83), https://doi.org/10.5281/zenodo.15097644 (ref. 84), https://doi.org/10.5281/zenodo.15212007 (ref. 71)) under the GNU General Public License v3.0. Other software used in this study are included in the Methods.

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

## Acknowledgements

We are grateful to M. Sachs for seed advice. We thank M. Krist, F. Eikelmann and F. Esser for field support. We thank S.-D. Pophaly, J. Reza and M. H. Thoben for IT support. We thank C. Becker, K. Becker and B. Huettel for sequencing support. We are grateful to S. Weizel, M. Blank, M. D. Mueller, R. S. Kumar, G. Koentjes, I. Tol, S. Mukherjee, N. Sargheini, L. Watschke, V. Rajan, E. van Moore and M. van Boesschoten for greenhouse and experimental support. We would like to thank F. Andrews for not irrigating our drought experiment. We also thank N. Springer for his critical discussion of the paper. This study was supported by funding from the 'Deutsche Forschungsgemeinschaft' (DFG, ID:458854361 to T.H.) as part of DFG Sequencing call 3 and the European Union's Horizon Europe program BOOSTER (ID: 101081770 to T.H.). B.S., W.B.F. and T.H. received funding from the DFG under Germany's Excellence Strategy (CEPLAS Cluster, EXC 2048/1, Project ID: 390686111). W.B.F. was supported by an Alexander von Humboldt professorship. J.R.I. was supported by funding from the United States National Science Foundation (ID: 1822330) and the United States Department of Agriculture (Hatch project CA-D-PLS-2066-H 548). J.C.S. and J.V.T.R. were supported by the United States Department of Energy (ID: DE-SC0020355). S.J.S. was supported by funding from the United States National Science Foundation through the Graduate Research Fellowship Program (DGE: 1744592). A.S. and H.W.B. were supported by the National Science Foundation Plant Genome Research Program (IOS 1444532). M. Stam was supported by European Commission Seventh Framework-People-2012-ITN Project EpiTRAITS, GA-316965 (Epigenetic Regulation of Economically Important Plant Traits). F.F. received support from the Program Oriented Funding of the Helmholtz Association. Most of the next-generation sequencing was performed and supported by the DFG Research Infrastructure West German Genome Center (407493903) as part of the Next Generation Sequencing Competence Network (project 423957469). Some of the results reported in this paper were partially supported by the HPC@ISU equipment at Iowa State University, some of which has been purchased through funding provided by NSF under MRI grant numbers 1726447 and MRI2018594. Some of the work done by J.E. was supported by the French National Research Agency (ANR, ID: ANR-22-CPJ2-0110-01). The funders had no role in study design, data collection and analysis, decision to publish or preparation of the paper.

## Author contributions

T.H. conceived the research project. J.E., S.J.S., J.R.I., D.E.R., M.B.H., B.S., M. Stam, W.B.F. and T.H. designed experiments. J.E., S.J.S., T.K., V.A.S.-C., R.B., A.S., J.V.T.R., A.K., D.T.H.D. and T.H. performed experiments. J.E., S.J.S., A.S., A.K., M-K.B., M. Schneider, J.R.I., D.E.R., G.S., V.A.S.-C., F.F., A.S.S., J.V.T.R., J.C.S., M.B. and T.H. analyzed the data. J.E., S.J.S., A.S.S. and T.H. developed the hybrid mapping pipeline and J.E., S.J.S., J.R.I., D.E.R., J.V.T.R., S.J.S., M. Schneider, A.K., M.B. and T.H. developed the analysis scripts. S.B. developed the custom genome browser. J.E., J.R.I. and T.H. wrote the paper and W.B.F., H.W.B., F.F., S.J.S. and B.S. revised it.

## FundingInformation

## Competing interests

The authors declare no competing interests.

## Additional information

**Extended data** is available for this paper at https://doi.org/10.1038/s41588-025-02246-7.

**Correspondence and requests for materials** should be addressed to Thomas Hartwig.

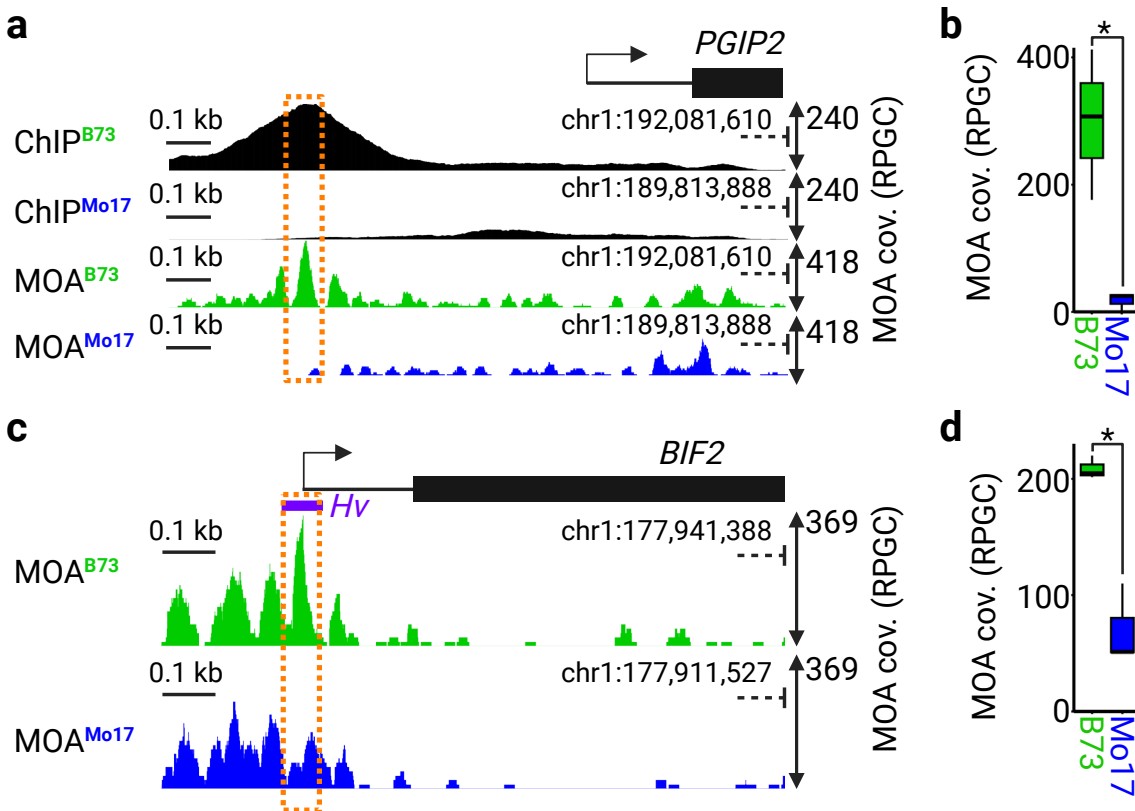

**Extended Data Fig. 1 | Examples of allele-specific B73xMo17 MOA-seq and comparison to allele-specific ChIP-seq. a)** Average, normalized, allele-specific ChIP-seq of the ZmBZR1 TF (top two rows, black) and MOA-seq (bottom two rows, B73 green, Mo17 blue; fragment center data, see methods) shown upstream of *ZmPGIP2*. **c)** Average, normalized, allele-specific differences in MOA coverage upstream of *ZmBIF2* overlap with a known, 'hypervariable' (Hv, purple box) *cis*-regulatory region. **b, d)** Normalized, average MOA coverage of three biological replicas at orange dashed boxes in **a** (P = 0.015, two-sided t-test) and **c** (P = 0.003, two-sided t-test). RPGC = number of reads per 1 bp / scaling factor (total number of mapped reads multiplied by fragment length / effective genome size). Boxes in plots denote the range from the first to the third quartile, and lines within boxes indicate the median. Whiskers represent 1.5-fold of the interquartile range.

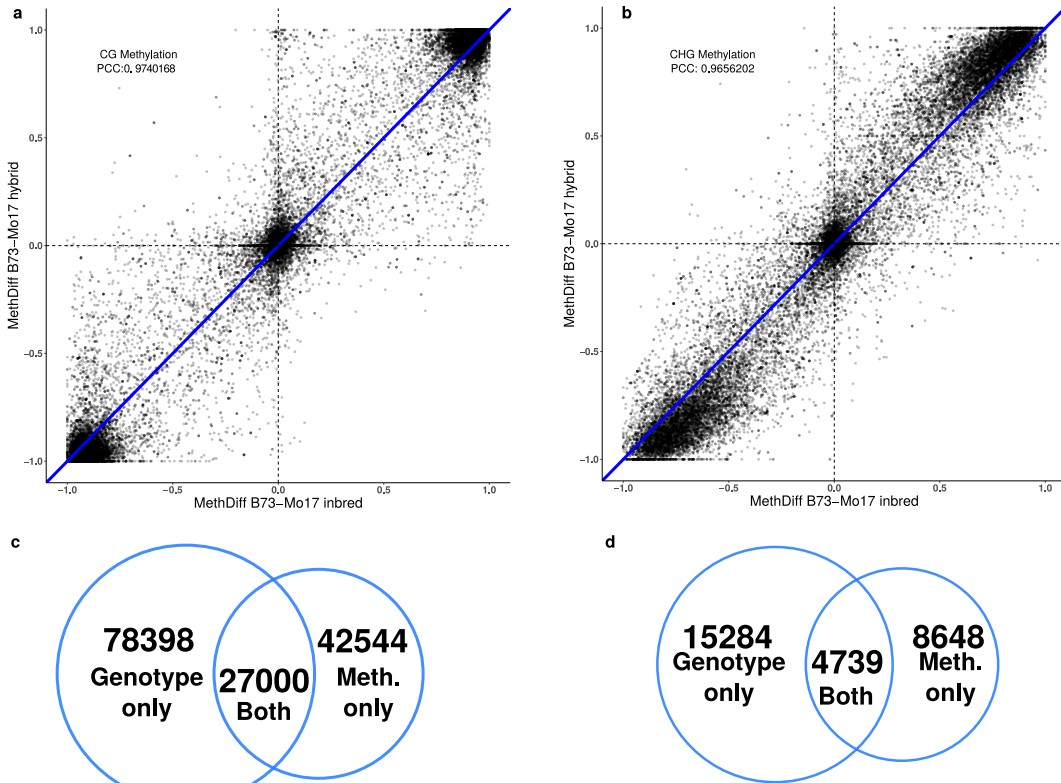

**Extended Data Fig. 2 | Correlation of methylation differences between the alleles in the B73xMo17 hybrid and the respective inbred parents and additional MOA-seq variation explained by methylation. a** and **b**) Methylation differences at AMP loci are expressed as Mo17 methylation values subtracted from B73 methylation values in inbred and hybrid comparisons (**a**: CG, **b**: CHG).

Blue line indicates x = y for orientation. PCC: Pearson Correlation Coefficient. **c** and **d**) Numbers of clumbed bQTL (SNP approach **c**, INDEL approach **d**) in WW conditions that display either a significant correlation of MOA-seq signal and the genotype (Genotype only), of MOA-seq signal and the methylation status within +/− 20 bp of the position (Meth. only) or both.

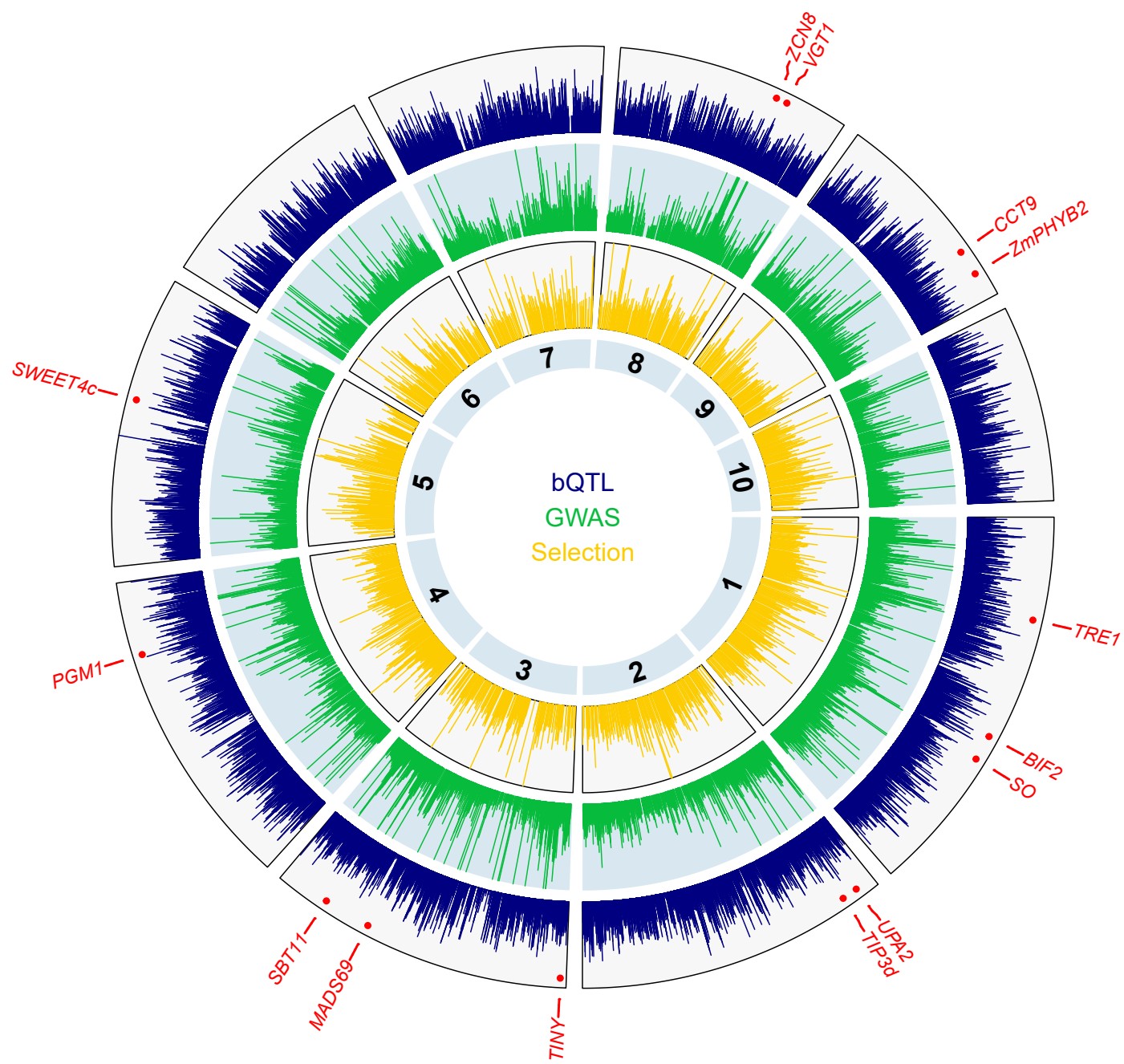

**Extended Data Fig. 3 | Distributions of bQTL, GWAS hits, and selective sweep genomic features across ten maize chromosomes.** From inner to outer circle, the tracks are: 1. chromosome names; 2. XP-CLR scores of selective sweeps[31] detected between modern maize and teosinte, 3. -log10 p-values of the maize GWAS Atlas lead SNPs[41]; and 4. -log10 p-values of bQTL (SNP and INDEL) associated with variation in MOA coverage in the pan-cistrome. Red markers denote selected examples of bQTL that coincide with natural variation of classical domestication and flowering time genes.

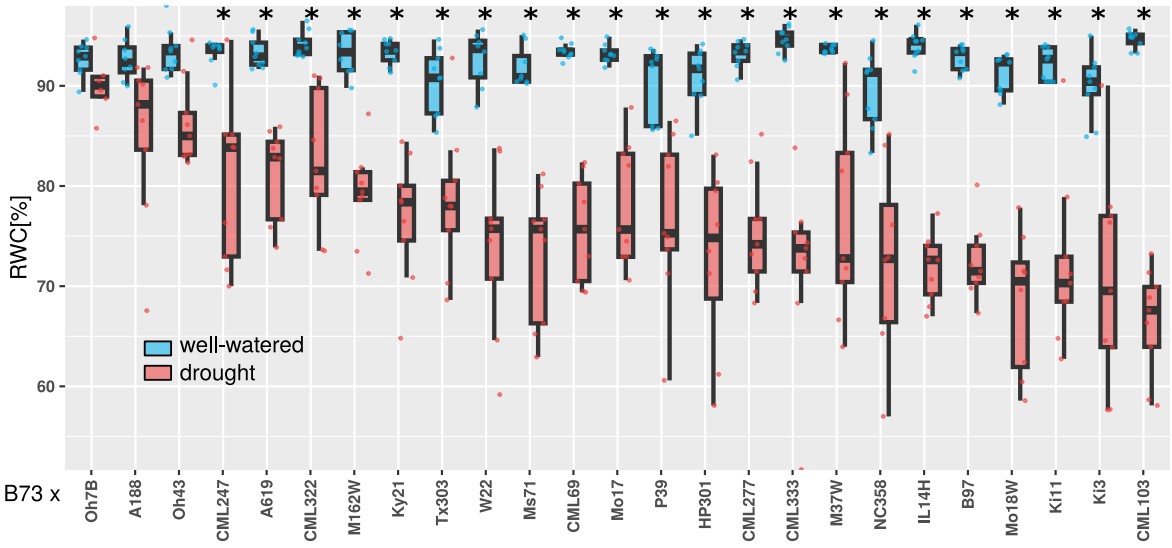

**Extended Data Fig. 4 | Assessment of relative water content of the 25 F1 hybrids under well-watered and drought conditions.** * indicate significant difference between WW and DS based on ANOVA followed by Tukey HSD test, p < 0.01 (exact p-values provided in source data). n = 9 pots with 4 plants each.

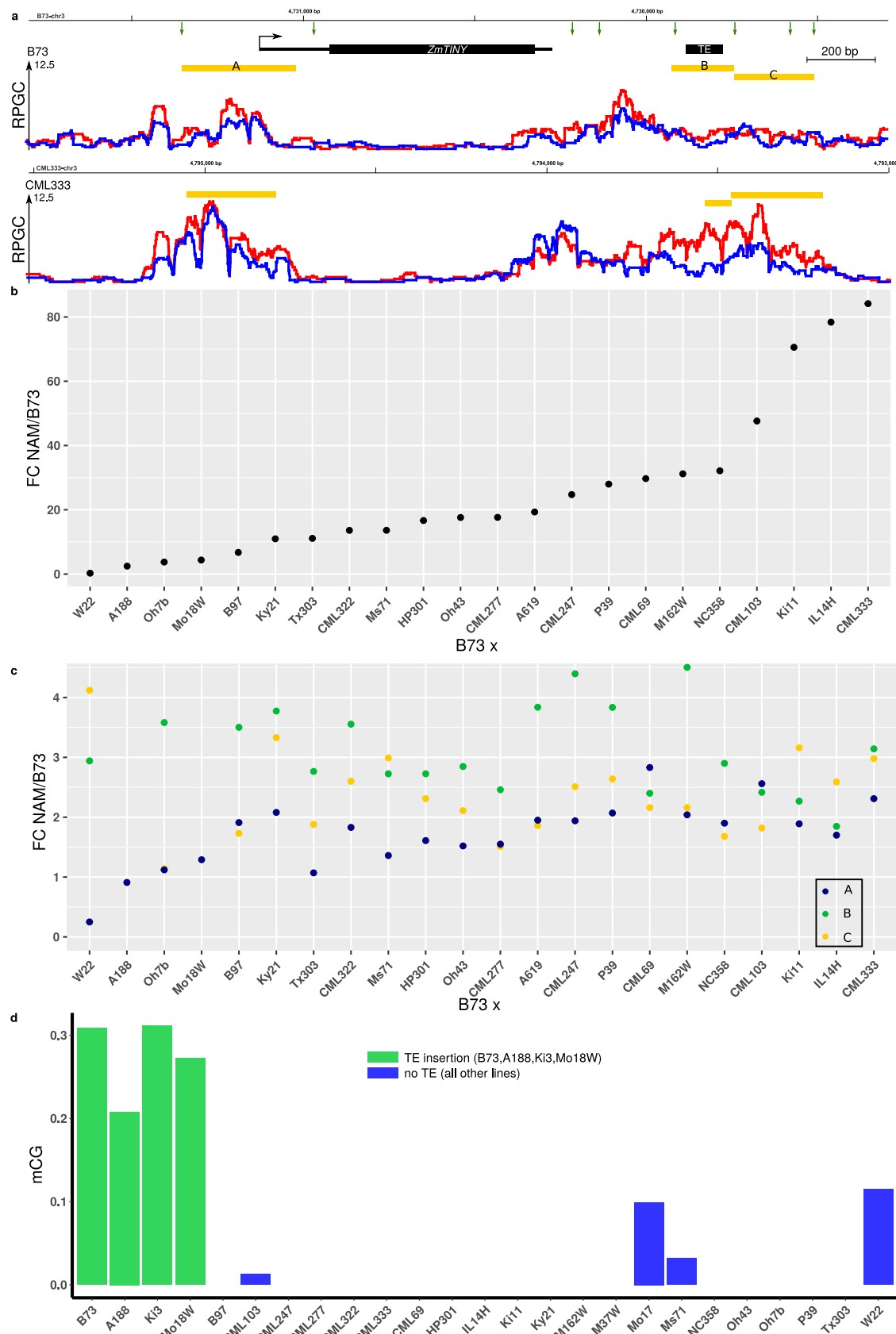

**Extended Data Fig. 5 | See next page for caption.**

**Extended Data Fig. 5 | Overview of the *ZmTINY* locus with allele-specific mRNA abundance and TF-binding. a**) Genome browser view of the upstream and downstream region of *ZmTINY* in the hybrid of B73xCML333. Green arrows mark DS-bQTL positions. Yellow blocks mark regions analyzed in (**c**). The black transposable elements (TE) block marks a TE that is present in B73, A188, Ki3, and Mo18W. The scale bar applies to both tracks. **b**) Fold change (FC) in mRNA abundance between the NAM allele and the B73 allele under DS conditions. **c**) FC in MOA occupancy between the NAM allele and the B73 allele under DS conditions. Average MOA occupancy per base was calculated in the three yellow regions marked in (**a**). To ensure comparison of homologous regions, regions between two homologous SNPs determined by whole-genome alignment were chosen. Note that the W22 promoter contains a TE in the (A) region, adding ~5 kb compared to the syntenic W22 region. Two additional regions in W22 were compared to the B73 (A) region: the region up to the TE (FC 0.36) and a region of similar length to the B73 (A) region (FC 0.21). The downstream regions of A188 and Mo18W do not differ from B73, and thus, no FC could be detected. **d**: CG methylation in 40 bp surrounding SNP B73-chr3:4729798 (just after the TE sequence downstream of *ZmTINY*).

# Reporting Summary

## Statistics

For all statistical analyses, confirm that the following items are present in the figure legend, table legend, main text, or Methods section.

| n/a | Confirmed | |
|---|---|---|
| ☐ | ☒ | The exact sample size (*n*) for each experimental group/condition, given as a discrete number and unit of measurement |
| ☐ | ☒ | A statement on whether measurements were taken from distinct samples or whether the same sample was measured repeatedly |
| ☐ | ☒ | The statistical test(s) used AND whether they are one- or two-sided<br>*Only common tests should be described solely by name; describe more complex techniques in the Methods section.* |
| ☐ | ☒ | A description of all covariates tested |
| ☐ | ☒ | A description of any assumptions or corrections, such as tests of normality and adjustment for multiple comparisons |
| ☐ | ☒ | A full description of the statistical parameters including central tendency (e.g. means) or other basic estimates (e.g. regression coefficient) AND variation (e.g. standard deviation) or associated estimates of uncertainty (e.g. confidence intervals) |
| ☐ | ☒ | For null hypothesis testing, the test statistic (e.g. $F$, $t$, $r$) with confidence intervals, effect sizes, degrees of freedom and $P$ value noted<br>*Give P values as exact values whenever suitable.* |
| ☒ | ☐ | For Bayesian analysis, information on the choice of priors and Markov chain Monte Carlo settings |
| ☒ | ☐ | For hierarchical and complex designs, identification of the appropriate level for tests and full reporting of outcomes |
| ☐ | ☒ | Estimates of effect sizes (e.g. Cohen's *d*, Pearson's *r*), indicating how they were calculated |

*Our web collection on statistics for biologists contains articles on many of the points above.*

## Software and code

Policy information about availability of computer code

| Data collection | Standard Illumina sequencing/demultiplexing pipeline |
|---|---|
| Data analysis | SeqPurge (v2022-07-15)<br>NGmerge (v0.3)<br>STAR (v2.7.7a)<br>SAMtools (v1.9)<br>unique-kmers.py (https://github.com/dib-lab/khmer/, commit fb65d21)<br>GNU Awk (v4.2.1)<br>Bedtools (v2.29.0 and v2.30.0)<br>bedGraphToBigWig (v 4)<br>seqtk trimfq (v1.3 r106)<br>trimmomatic (v0.39)<br>DeepTools (v3.5.0 and v.3.5.5)<br>bwa-mem2 (v2.2.1)<br>GATK (v4.3.0.0)<br>G2Gtools (v. 0.2.7)<br>progressive cactus (v1.0.0 2020-04-19)<br>hallLiftover (hal-release-v2.1)<br>MACS3 (v3.0.1)<br>Anchorwave (v1.2.2)<br>minimap2 (2.27-r1193) |

CrossMap (version 0.6.4 and 0.7.0)
wgatools (version 0.1.0)
Julia (1.8.1)
R (4.1.1, 4.1.2, and 4.4.2)
Bismark (v0.22.3)
bowtie2 (v2.4.4)
LDAK (v5.2)
Tassel (v5)
CrossMap ( v0.6.4)

GitHub Repositories:
https://github.com/Snodgras/MOA_Analysis
https://github.com/corn2code/bQTL
https://github.com/jengelhorn/AS-MOA
https://github.com/jengelhorn/AS-RNAseq
https://github.com/Ako31415/FIND-CIS-analysis

For manuscripts utilizing custom algorithms or software that are central to the research but not yet described in published literature, software must be made available to editors and reviewers. We strongly encourage code deposition in a community repository (e.g. GitHub). See the Nature Portfolio guidelines for submitting code & software for further information.

## Data

Policy information about availability of data

All manuscripts must include a data availability statement. This statement should provide the following information, where applicable:
- Accession codes, unique identifiers, or web links for publicly available datasets
- A description of any restrictions on data availability
- For clinical datasets or third party data, please ensure that the statement adheres to our policy

All MOA-seq and RNA-seq raw data generated for this publication have been deposited at NCBI SRA under accession number PRJNA1101486. MOA coverage tacks and peak files have been deposited on Gene Expression Omnibus under GSE294039 and will also be included in future releases at MAIZEDGB. Coverage and binding frequency data for all bQTL is accessible at a custom browser at: https://www.plabipd.de/ceplas/?config=maize_hartwig_config.json and zendono (https://doi.org/10.5281/zenodo.15177272). For convenience concatenated genomes were also deposited at zenodo (https://doi.org/10.5281/zenodo.15177272).

## Research involving human participants, their data, or biological material

Policy information about studies with human participants or human data. See also policy information about sex, gender (identity/presentation), and sexual orientation and race, ethnicity and racism.

| Reporting on sex and gender | n/a |
| --- | --- |
| Reporting on race, ethnicity, or other socially relevant groupings | n/a |
| Population characteristics | n/a |
| Recruitment | n/a |
| Ethics oversight | n/a |

Note that full information on the approval of the study protocol must also be provided in the manuscript.

# Field-specific reporting

Please select the one below that is the best fit for your research. If you are not sure, read the appropriate sections before making your selection.

☒ Life sciences ☐ Behavioural & social sciences ☐ Ecological, evolutionary & environmental sciences

For a reference copy of the document with all sections, see nature.com/documents/nr-reporting-summary-flat.pdf

# Life sciences study design

All studies must disclose on these points even when the disclosure is negative.

| Sample size | We proposed to construct a first-generation pan-cistrome for maize. To do that, we focused on diverse genotypes with high-quality genome assemblies available at the time of the study, that could be crossed under field conditions in Germany. Our results indicate that 25 F1 lines are sufficient for the association analyses and that our population was near saturation (Fig. 2, Supplementary Fig. 6). Four F1 or inbred plants per pot and 3 pots (12 plants total) per treatment and per replicate were chosen. No statistical methods were used to pre-determine sample sizes but our sample sizes are similar to those reported in previous publications (Hartwig et al., 2023, doi:10.1186/s13059-023-02909-w). |
| --- | --- |

| | |
|---|---|
| Data exclusions | no data was excluded |
| Replication | All experiments unless otherwise stated in the Method section were performed with three biological replicates. Plants were grown at separate time points for the three biological replicate. All attempts of replication were successful. |
| Randomization | All plants were grown in a randomized block design. At the start of the treatment all plants were re-randomized and grown in a randomized block design during the treatment. Harvest order was randomized per each harvest, except that well-watered and drought samples for each line were harvested consecutively to minimize differences. |
| Blinding | Blinding was not required as there are no participant biases involved in this plant molecular genetics and genomics work. |

# Reporting for specific materials, systems and methods

We require information from authors about some types of materials, experimental systems and methods used in many studies. Here, indicate whether each material, system or method listed is relevant to your study. If you are not sure if a list item applies to your research, read the appropriate section before selecting a response.

## Materials & experimental systems

| n/a | Involved in the study |
|---|---|
| ☒ | ☐ Antibodies |
| ☒ | ☐ Eukaryotic cell lines |
| ☒ | ☐ Palaeontology and archaeology |
| ☒ | ☐ Animals and other organisms |
| ☒ | ☐ Clinical data |
| ☒ | ☐ Dual use research of concern |
| ☐ | ☒ Plants |

## Methods

| n/a | Involved in the study |
|---|---|
| ☒ | ☐ ChIP-seq |
| ☒ | ☐ Flow cytometry |
| ☒ | ☐ MRI-based neuroimaging |

## Dual use research of concern

Policy information about dual use research of concern

### Hazards

Could the accidental, deliberate or reckless misuse of agents or technologies generated in the work, or the application of information presented in the manuscript, pose a threat to:

| No | Yes |
|---|---|
| ☒ | ☐ Public health |
| ☒ | ☐ National security |
| ☒ | ☐ Crops and/or livestock |
| ☒ | ☐ Ecosystems |
| ☒ | ☐ Any other significant area |

### Experiments of concern

Does the work involve any of these experiments of concern:

| No | Yes |
|---|---|
| ☒ | ☐ Demonstrate how to render a vaccine ineffective |
| ☒ | ☐ Confer resistance to therapeutically useful antibiotics or antiviral agents |
| ☒ | ☐ Enhance the virulence of a pathogen or render a nonpathogen virulent |
| ☒ | ☐ Increase transmissibility of a pathogen |
| ☒ | ☐ Alter the host range of a pathogen |
| ☒ | ☐ Enable evasion of diagnostic/detection modalities |
| ☒ | ☐ Enable the weaponization of a biological agent or toxin |
| ☒ | ☐ Any other potentially harmful combination of experiments and agents |

## Plants

| | |
|---|---|
| Seed stocks | B73, Mo17, A619, W23, W22, A188, and US-NAM seeds were supplied by the GRIN National Agricultural Library. |
| Novel plant genotypes | n/a |
| Authentication | In addition to GRIN documentation, whole genome alignment was performed and sequence variants validated in the sequiencing data to ensure genotype authenticity |

