## [Peer Review File · Nature Genetics]

Genetic variation at transcription factor binding sites largely explains phenotypic heritability in maize

Corresponding Author: Dr Thomas Hartwig

Version 0:

Decision Letter:

19th Jun 2024

Dear Dr Hartwig,

Your Article, "Genetic variation at transcription factor binding sites largely explains phenotypic heritability in maize" has now been seen by 3 referees. You will see from their comments below that while they find your work of interest, some important points are raised. We are interested in the possibility of publishing your study in Nature Genetics, but would like to consider your response to these concerns in the form of a revised manuscript before we make a final decision on publication.

To guide the scope of the revisions, the editors discuss the referee reports in detail within the team, including with the chief editor, with a view to identifying key priorities that should be addressed in revision and sometimes overruling referee requests that are deemed beyond the scope of the current study. In this case, we invite you to address Reviewers' comments in full. We hope that you will find the prioritized set of referee points to be useful when revising your study. Please do not hesitate to get in touch if you would like to discuss these issues further.

We therefore invite you to revise your manuscript taking into account all reviewer and editor comments. Please highlight all changes in the manuscript text file. At this stage we will need you to upload a copy of the manuscript in MS Word .docx or similar editable format.

*2) If you have not done so already please begin to revise your manuscript so that it conforms to our Article format instructions, available

[here](http://www.nature.com/ng/authors/article_types/index.html).

*3) Include a revised version of any required Reporting Summary: <https://www.nature.com/documents/nr-reporting-summary.pdf>

Please be aware of our [guidelines](https://www.nature.com/nature-research/editorial-policies/image-integrity) on digital image standards.

Link Redacted

We hope to receive your revised manuscript within four to eight weeks. If you cannot send it within this time, please let us know.

Nature Genetics is committed to improving transparency in authorship. As part of our efforts in this direction, we are now requesting that all authors identified as 'corresponding author' on published papers create and link their Open Researcher and Contributor Identifier (ORCID) with their account on the Manuscript Tracking System (MTS), prior to acceptance. ORCID helps the scientific community achieve unambiguous attribution of all scholarly contributions. You can create and link your ORCID from the home page of the MTS by clicking on 'Modify my Springer Nature account'. For more information please visit please visit www.springernature.com/orcid.

Sincerely,
Chiara

Chiara Anania, PhD
Associate Editor
Nature Genetics
<https://orcid.org/0000-0003-1549-4157>

Referee expertise:

Referee #1: plant genomics/population genetics

Referee #2: plant molecular breeding

Referee #3: plant transcription/epigenetics

Reviewers' Comments:

Reviewer #1:

Remarks to the Author:

Engelhorn et al. constructed a pan-cistrome of the maize leaf under watered and drought conditions with MNase-defined cistrome Occupancy Analysis (MOA-seq), and therefore quantified globally haplotype-specific TF footprints across a pan-genome of 25 maize hybrids and mapped over 210,000 genetic variants (termed binding-QTL) linked to cis-element occupancy with high resolution. The authors also provided strong evidence to support the functional binding-QTL. This work provides a paradigm for identifying genome-wide, high-resolution maps of functional variants in cis-elements. The work is interesting, the data are solid, and has potential in application, however, several issues and major concerns need to be addressed.

1. For potential application: MOA-seq enables identification of transcription factor binding sites at a high resolution (~65 bp fragments according to the article). Additionally, considering that some reported functional variations disrupting transcription factor binding sites are of the INDEL polymorphisms, is there any potential or possible to directly identify functional variations related to cis-regulatory elements by using MOA INDELS as markers in bQTL association analysis? This point should be addressed in the manuscript.

2. For a concern: As indicated by the manuscript, differences in DNA methylation can affect TF binding affinity. Two NAM parents, Ki3 and CML69, harbor a PIF/Harbinger transposon upstream of ZmPGM1, resulting in hypermethylation of the DNA between MOA peak and the TSS, a pattern not observed for alleles without the transposon. Thus, it is puzzling that these two NAM parents show similar MOA coverage to B73, which does not contain the transposon (Fig. 2i-k). This concern should be addressed as well.

3. A suggestion: The transcription levels of the gene ZmRAP2.7 should be compared under the regulation of the two VGT1-MOA haplotypes to validate the function of the cis-regulatory element VGT1-MOA.

4. A correction: The citation of Fig. 4d-f in the article does not match the figure.

Reviewer #2:

Remarks to the Author:

The authors present a high-throughput method for identifying functional, (epi-)genetic variants linked to trait variation in plants. Using this approach, they quantified haplotype-specific TF footprints across a pan-genome of 25 maize hybrids and mapped over two-hundred thousand binding-QTL linked to cis-element occupancy. These binding-QTLs capture the majority of heritable trait variation across ~70% of 143 phenotypes. The study is efficient in revealing hidden cis-variation for genetic studies and multi-target engineering of complex traits. Therefore, the method is valuable for helping researchers pinpoint casual sequence variations, such as SNPs, responsible for phenotypic diversity. Overall, the data is very interesting. The manuscript can be improved with revisions.

Major concerns:

The establishment of a given phenotype usually involves numerous genes, with only those genes that exhibit natural variations contributing to phenotypic diversity. Whereas, genes exhibiting no functional variations would not impact phenotypic diversity. Using the MOA-seq method, a huge number of MPs have been identified, which actually saturate all functional genes. This suggests that most MPs or AMPs are not actually responsible for phenotypic diversity. Furthermore, due to the difference in genetic backgrounds between the two parental lines, most trans-acting factors (mostly transcription factors) must show allelic variations and bind preferentially to certain cis-elements. However, based on the current method, the authors could not distinguish allelic variations for these trans-acting factors. Thus, I question the conclusion that the high concordance between F1 and inbred MOA-seq results from the coupling of the majority of AMPs with genotypic differences in cis at the binding site, rather than from trans-acting or cis-by-trans interaction effects.

The authors used many genes to confirm the usefulness of their approach. The most important issue is that, before detecting bQTL between two parental lines, the authors need to provide concrete evidence that the difference in a certain trait between two lines results from the bQTL, rather than other genetic factors across the genome. Additionally, most figures and their legends need improvement.

Other comments:

Original statement in Abstract: "Here we report the construction of a pan-cistrome of the maize leaf under well-watered and drought conditions" is overstated. Throughout the manuscript, only the last experiment was conducted under well-watered and drought conditions.

The original statement: "The high concordance between F1 and inbred MOA-seq further establishes the reproducibility of the assay and indicates that the majority of AMPs are coupled to genotypic differences in cis at the binding site, rather than resulting from trans-acting or cis-by-trans interaction effects," is premature. Since the allelic differences in trans-acting factors between the two parental lines are unknown, it is too early to draw this conclusion. Additionally, SNPs in bQTLs may not be located in the binding motif and therefore may not function as cis-elements.

The original sentence: "The MOA-seq AMP footprint overlaps a small 80 bp hypervariable region in the BIF2 proximal promoter (in Fig. 1g) previously associated with differences between B73 and Mo17 in traits such as tassel branch zone length, plant height, and leaf width and length (Pressoir et al. 2009)". Comment: Is this 80 bp hypervariable region the causal factor for the phenotypic difference between B73 and Mo17? Further evidence is needed to confirm this causal relationship.

The original sentence: "We next sought to identify (epi)genetic variants associated with differences in MOA-detected TF occupancy between haplotypes, or binding quantitative trait loci (bQTL), in our population". Comment: I would like to investigate histone methylation profiles, which are more closely related to chromatin structure.

The original statement: "SNPs alone explained 45% of the observed differences in MOA coverage, whereas SNPs and DNA methylation combined explained 57%". Comment: this suggests that DNA methylation does not contribute much to the observed differences in MOA coverage. This result further indicates that histone methylation may be more influential than DNA methylation.

The original statement: "As expected the genome-wide distribution of bQTL was distinct from all SNPs and more closely matched those of TF target sites (Tu et al. 2020) and allele-specific ZmBZR1 binding sites (Additional file 1, Fig. S8, (Hartwig et al. 2023))". Comment: This is probably due to linkage disequilibrium.

The original sentence: "...the causative transposon insertions at ZmCCT9 and ZmCCT10 (Huang et al. 2018)". Comment: the authors need to cite the relevant article. The transposon insertion at ZmCCT10 was not originally reported in this article.

The original sentence: "We observed haplotype-specific footprints, both at the previously reported 8 bp insertion (Wen et al. 2018) and an additional SNP 29 bp upstream, which coincided with a bQTL". Comment: are these sequence variations also responsible for the trait difference between B73 and Mo17?

The original sentence: "whereas F1s with no polymorphism between B73 and NAM showed no difference in the haplotype-specific transcript levels (Fig. 2i-k)". Comment: This is because the lack of polymorphism failed to distinguish between B73 and NAM, but it does not indicate there is no difference in haplotype-specific transcript levels. For example, the difference in DNA or histone methylation profile resulting from distal transposon insertion must impact haplotype-specific transcript levels.

The original statement: "MOA signals in the TINYB73 and TINYCML333 upstream promoters showed the highest correlation to ZmTINY mRNA levels (Additional File 1, Figure S22)". Comment: are these MOA signals responsible for natural variation at TINYB73 and TINYCML333

The original statement: "Finally, we highlight the relevance of bQTL loci for understanding phenotypic diversity in maize,

demonstrating that haplotype-specific MOA-seq allowed us to capture the majority of additive genetic variation for most tested phenotypes in maize leaves". Comment: only the bQTL related to natural variations allowed us to capture the additive genetic variation for most tested phenotypes. However, most bQTL that function in leaf development are not related to genetic variation.

Fig. 1c, d: These figures are hard to understand and need modification for better clarity.

"Fig. 1g,h" In panel g, I do not find much difference in MOA coverage between B73 and Mo17; however, in panel h, MOA coverage in B73 is much higher than that of Mo17.

Reviewer #3:

Remarks to the Author:

Traditional experimental methods, such as TF ChIP-seq and DNase-seq (including ATAC-seq or MOA-seq), have limitations and drawbacks when it comes to find cis-regulatory elements (CREs). On the other hand, computationally inferring CREs from GWAS or other population genetics-based approaches is prone to false positives, as these methods can identify too many SNPs that are merely linked to the true CRE.

In this study, the authors cleverly used an F1 hybrid approach to overcome the false positive problem. By analyzing the ratio of the two parent's SNPs within the transcription factor (TF) binding sites, they were able to infer whether a particular SNP was actually affecting TF binding, and thus identify the true CREs or at least a small region with multiple SNPs that contain the true CRE. An additional advantage of their approach is that it does not require knowing which specific TF can bind to the identified sites. The authors used MOA-seq, which can broadly identify all TF-protected regions across the genome, without the need to examine each individual TF. Also, the MOA-seq is much easier to performed compared to the DNase-seq.

They first tested MOA-seq on the two parental lines, B73 and Mo17, as well as their F1 hybrid. By leveraging their previously published allelic ZmBZR1 binding data for those same lines, they were able to demonstrate that the allelic bias observed in their MOA-seq peaks closely correlated with the allelic patterns of TF binding.

Next, the authors expanded this approach to 25 F1 lines from the NAM population. They reported that the binding QTLs identified through this method could explain 101 out of 143 phenotypic traits in the NAM population, outperforming traditional GWAS (Genome-Wide Association Study) approaches.

Finally, the authors subjected their 25 F1 lines to drought treatment and used the same method to identify cis-regulatory elements (CREs) that are responsive to drought conditions.

While the authors' approach of using F1 hybrids to identify true cis-regulatory elements (CREs) is clever and effective, they should explain to the readers that there is a big limitation. The success of this strategy is dependent on the F1 population. Specifically, the F1 hybrids must have sufficient heterozygosity within the TF binding regions for the method to work effectively. This could be a significant drawback, as other plants, such as arabidopsis, rice and tomato, tend to have much lower levels of heterozygosity compared to maize. So this method is probably only suitable for maize. In addition, this method will miss all the traits that requires a homozygous locus.

For the title, will it be more accurate to claim that it can "explains phenotypic heritability in maize F1 hybrid"?

The author frequently used the terms "TF binding footprint" and "cis-regulatory elements (CRE)". Footprint and CRE typically refer to a DNA motif that is bound by the TF, which is ~ 10 bp, while their MOA-seq peak is about 50-100bp? It is better to define it better to help the readers to understand the difference.

The penultimate result section compares their method with others. I think this is an important part. But it is quite hard to read. Background SNP account for more genetic variation? Why? How do they match the distance to nearest gene? Are they in the open chromatin? Using whole-genome's SNP can't get similar result? Will it be the same to use allelic SNP in ATAC-seq peaks? I hope the author can improve this part and let the reader understand what is the pros and cons of their method.

The method section lacks details to reproduce the analysis. I understand the space limit but they can and should put a detail method into the supplementary information. Also they should put their raw data to NCBI SRA and processed data to GEO. Currently, the link they provide are not accessible. So I can't check the quality of the data.

Version 1:

Decision Letter:

Our ref: NG-A65421R

31st Jan 2025

Dear Dr. Hartwig,

Thank you for submitting your revised manuscript "Genetic variation at transcription factor binding sites largely explains phenotypic heritability in maize" (NG-A65421R). It has now been seen by the original referees and their comments are below. The reviewers find that the paper has improved in revision, and therefore we'll be happy in principle to publish it in Nature Genetics, pending minor revisions to comply with our editorial and formatting guidelines.

Congratulations!

Sincerely,
Chiara

Chiara Anania, PhD
Associate Editor
Nature Genetics
<https://orcid.org/0000-0003-1549-4157>

Reviewer #2 (Remarks to the Author):

The authors address all my questions, so I have no more questions but recommend it to be accepted by NG.

Reviewer #3 (Remarks to the Author):

I have no further comments.

Reviewer 1

Remarks to the Author:

Engelhorn et al. constructed a pan-cistrome of the maize leaf under watered and drought conditions with MNase-defined cistrome Occupancy Analysis (MOA-seq), and therefore quantified globally haplotype-specific TF footprints across a pan-genome of 25 maize hybrids and mapped over 210,000 genetic variants (termed binding-QTL) linked to cis-element occupancy with high resolution. The authors also provided strong evidence to support the functional binding-QTL. This work provides a paradigm for identifying genome-wide, high-resolution maps of functional variants in cis-elements. The work is interesting, the data are solid, and has potential in application, however, several issues and major concerns need to be addressed.

1. For potential application: MOA-seq enables identification of transcription factor binding sites at a high resolution (~65 bp fragments according to the article). Additionally, considering that some reported functional variations disrupting transcription factor binding sites are of the INDEL polymorphisms, is there any potential or possible to directly identify functional variations related to cis-regulatory elements by using MOA INDELs as markers in bQTL association analysis? This point should be addressed in the manuscript.

We thank the reviewer for this suggestion. We agree that INDEL polymorphisms have the potential to affect TF binding and traits and as such INDEL-related bQTL should be included in our study. We modified our bQTL pipeline and re-analysed the bQTL data to now include INDELs as well. We identified a total of 28,671 and 23,554 INDEL bQTL in well-watered and DS conditions, respectively, for a total of 40,240 INDEL bQTL. These include INDELs at known cis-variations such as the hypervariable region in the BIF2 promoter. We included these findings in the manuscript. Overall, we find that INDEL variants show patterns very similar to SNPs, with e.g., 63% of WW INDEL bQTL overlapping with a SNP bQTL within 65bp. We think these results enhance the high-resolution map, but because of the substantial overlap we present most of our results with just the SNP bQTL, in line with recent findings that structural variants do not add a large fraction to explaining heritability (<https://www.biorxiv.org/content/10.1101/2024.06.14.599082v2.full.pdf>).

2. For a concern: As indicated by the manuscript, differences in DNA methylation can affect TF binding affinity. Two NAM parents, Ki3 and CML69, harbor a PIF/Harbinger transposon upstream of ZmPGM1, resulting in hypermethylation of the DNA between MOA peak and the TSS, a pattern not observed for alleles without the transposon. Thus, it is puzzling that these two NAM parents show similar MOA coverage to B73, which does not contain the transposon (Fig. 2i-k). This concern should be addressed as well.

We thank the reviewer for pointing out that the initial phrasing of the sentence could be misleading. The TE Insertion site in Ki3 and CML69 is located outside the MOA footprint peak. The MOA peak is not directly affected by the TE and does not show a change in DNA methylation, consistent with no observed change in MOA coverage. Only Ki3 and CML69, but none of the other lines including B73, show a TE insertion and hypermethylated site between the MOA peak site and the TSS. We modified the sentence to clarify this: Two NAM parents, Ki3 and CML69, showed much lower PGM1 transcript levels, while no significant variation in MOA footprint was detected. Instead Ki3 and CML69 harbored a PIF/Harbinger transposon

insertion accompanied by hypermethylation between the MOA peak and PGM1 TSS, not found in any of the haplotypes (including B73) with higher PGM1 transcript levels.

3. A suggestion: The transcription levels of the gene *ZmRAP2.7* should be compared under the regulation of the two VGT1-MOA haplotypes to validate the function of the cis-regulatory element VGT1-MOA.

We did analyse this suggestion. We compared the VGT-MOA haplotypes at the bQTL and *RAP2.7* transcript level and did not observe a clear correlation. This is to be expected as both *vgt1* and *vgt1-DMR* are also known to control *RAP2.7* transcripts and given the large distance between them, all three loci are not linked. As a result, we observed various combinations of the three loci making it impossible with only 25 lines to relate their genotype combinations to *RAP2.7* transcripts. We added the statement: "However, future functional tests are needed to establish whether VGT1-MOA effects *ZmRAP2.7* expression alone or in combination with VGT1 and/or VGT1-DMR."

4. A correction: The citation of Fig. 4d-f in the article does not match the figure.

Thank you for picking up this error. We corrected it in the text.

Reviewer #2:

Remarks to the Author:

The authors present a high-throughput method for identifying functional, (epi-)genetic variants linked to trait variation in plants. Using this approach, they quantified haplotype-specific TF footprints across a pan-genome of 25 maize hybrids and mapped over two-hundred thousand binding-QTL linked to cis-element occupancy. These binding-QTLs capture the majority of heritable trait variation across ~70% of 143 phenotypes. The study is efficient in revealing hidden cis-variation for genetic studies and multi-target engineering of complex traits. Therefore, the method is valuable for helping researchers pinpoint casual sequence variations, such as SNPs, responsible for phenotypic diversity. Overall, the data is very interesting. The manuscript can be improved with revisions.

Major concerns:

The establishment of a given phenotype usually involves numerous genes, with only those genes that exhibit natural variations contributing to phenotypic diversity. Whereas, genes exhibiting no functional variations would not impact phenotypic diversity. Using the MOA-seq method, a huge number of MPs have been identified, which actually saturate all functional genes. This suggests that most MPs or AMPs are not actually responsible for phenotypic diversity.

The reviewer is indeed correct that our approach will miss causal loci lacking variation in our study. This is of course a limitation of any experiment studying phenotypic diversity (QTL mapping, GWAS, eQTL, etc). We do indeed find AMPs in a large proportion of genes, although the number of AMPs present in any individual line is less than the total we find in all 25. The fact that we find many AMPs does not, however, mean that they are not responsible for phenotypic diversity. Indeed, while many AMPs may have small or no effects on some traits, the fact that we can explain the majority of phenotypic variation with our bQTLs, which explain

much less than 1% of the genome, provides quantitative support to the importance of these regions. Moreover, we show individual examples for a number of genes, for example *ZmTINY*, where MOA variation directly impacts gene function. We are not claiming that all bQTLs affect all traits, of course, but the data strongly suggest that our MOA-seq approach successfully identifies the regions of the genome most pertinent for phenotypic variation.

Furthermore, due to the difference in genetic backgrounds between the two parental lines, most trans-acting factors (mostly transcription factors) must show allelic variations and bind preferentially to certain cis-elements. However, based on the current method, the authors could not distinguish allelic variations for these trans-acting factors. Thus, I question the conclusion that the high concordance between F1 and inbred MOA-seq results from the coupling of the majority of AMPs with genotypic differences in cis at the binding site, rather than from trans-acting or cis-by-trans interaction effects.

Thank you for raising this question. In F1 hybrids there are no trans-effects due to the TFs itself. Since the comparison of the haplotypes is within the same cell, only cis-effects or cis-by-trans effects should be detected. Presence/absence variation of TFs, or the abundance of active TF which may differ between inbreds, or condition differences will not affect haplotype-specific occupancy in the same cell. That is why we first determined haplotype/specific variants in the F1 and only then compared the same loci detected in the F1 with inbred lines. Indeed our results show that the overwhelming majority of variants identified in the F1 show a similar differential bias in the inbred lines. This strongly supports that, as should be expected, the variants identified in our study are acting in cis or cis-by-trans. It is true that if we had compared differential binding in the inbred parents per se, it is likely that a substantial part of the differential occupancy may be due to trans-effects. We thank the reviewer for this remark and made this point clearer in the manuscript.

The authors used many genes to confirm the usefulness of their approach. The most important issue is that, before detecting bQTL between two parental lines, the authors need to provide concrete evidence that the difference in a certain trait between two lines results from the bQTL, rather than other genetic factors across the genome. Additionally, most figures and their legends need improvement.

Thank you for your comment. This is exactly the experiment we conducted with our VCAP approach. We model phenotypic variation in the NAM RILs as being caused by bQTL, matched background SNPs, and SNPs in the rest of the genome. Our simulations suggest this approach works well in identifying the relative contribution of each of these sets of SNPs – that is, when bQTL do not explain variation for a trait, our method assigns them a correspondingly low importance. Our results, however, show that, for more than 70% of the 143 phenotypes we evaluated, bQTL explains the majority of the genetic variation, even after accounting for segregating SNPs across the rest of the genome.

We apologize for difficulties with figures; we have updated several figure panels and added additional explanations in the legend sections to make the figures more clear.

Other comments:

Original statement in Abstract: “Here we report the construction of a pan-cistrome of the maize leaf under well-watered and drought conditions” is overstated. Throughout the manuscript, only the last experiment was conducted under well-watered and drought conditions.

Thank you for making us aware that this part of the experimental procedure needs clarification. We added the below statement to make it more clear that the entire haplotype-specific TF footprinting for all 25 F1 lines and association mapping to detected variants associated with haplotype-specific MOA occupancy was performed not only under WW but also DS condition. We hope this makes it clear that the pancistrome was constructed under both conditions to make the statement correct. While our descriptions of the WW pan-cistrome also included general features, we focused our analysis of the pan-cistrome under drought conditions on demonstrating the potential of our method to specific traits of interest.

We added part in the drought paragraph: To evaluate drought-induced differences in *cis*-element regulation, we performed haplotype-specific MOA- and RNA-seq in drought samples of all 25 F1 hybrids grown at the same time as the well-watered samples. We detected on average 287,844 MPs and 56,863 AMPs, slightly less than under well-watered conditions (Supplemental Table S5) and observed a similar correlation with allele-specific transcript abundance (Additional file 1, Fig. S12).

The original statement: “The high concordance between F1 and inbred MOA-seq further establishes the reproducibility of the assay and indicates that the majority of AMPs are coupled to genotypic differences in *cis* at the binding site, rather than resulting from trans-acting or cis-by-trans interaction effects,” is premature. Since the allelic differences in trans-acting factors between the two parental lines are unknown, it is too early to draw this conclusion. Additionally, SNPs in bQTLs may not be located in the binding motif and therefore may not function as *cis*-elements.

Thanks for this comment. As we mentioned for a point raised above, we amended our text to clarify that trans-factor differences that may exist between the inbred parents will not affect the relative haplotype-specific footprints when both genotypes are in the same cell. If all variation in binding were due to *cis*-effects alone, we would expect the F1 analysis to perfectly mirror what we see in inbreds. Differences between the F1 and inbreds would be caused by trans- or cis trans- acting effects that differ between the inbred parents (but are by definition the same for both alleles in an F1). Such trans-acting effects could lead to differences in binding between inbreds even when there are no sequence or epigenetic differences at the binding site. However, the fact that the F1 mostly (but not completely) mirrors the inbreds supports the expectation that most binding site variation is due to differences in *cis*, but the fact that this is not 100% of the time highlights the importance of looking for binding site differences in the F1, where we can rule out trans effects. We re-formulated the paragraph to make these points more clear.

The original sentence: “The MOA-seq AMP footprint overlaps a small 80 bp hypervariable region in the BIF2 proximal promoter (in Fig. 1g) previously associated with differences between B73 and Mo17 in traits such as tassel branch zone length, plant height, and leaf width and length (Pressoir et al. 2009)”. Comment: Is this 80 bp hypervariable region the causal factor for the phenotypic difference between B73 and Mo17? Further evidence is needed to confirm this causal relationship.

Thank you for pointing this out. We now added information about the *bif2* mutant phenotype, showing the dramatic effect of loss of *BIF2* function. Pressoir et al. collected several lines of evidence linking the genetic variation in the *BIF2* promoter to phenotypic differences: They detected complete linkage to traits, further conducted linkage mapping in IBM population and allele-specific expression analysis, showing the same trend for AS expression as we do with lower expression of *BIF2* from the Mo17 allele in F1 hybrids. They further performed RACE and could not find any evidence for alternative transcription start sites as an alternative explanation for the observed differences, nor could they identify any evidence that expression differences or phenotypic differences are due to variation in the coding region. Multiple lines of evidence thus suggesting this hypervariable site is causative. Our evidence supports these findings, we detect both a SNP bQTL and an INDEL bQTL inside this hypervariable region. Deciphering the role of all polymorphism within this region and their single contributions to allele-specific expression and phenotypic variation however, is beyond the scope of this story.

The original sentence: “We next sought to identify (epi)genetic variants associated with differences in MOA detected TF occupancy between haplotypes, or binding quantitative trait loci (bQTL), in our population”. Comment: I would like to investigate histone methylation profiles, which are more closely related to chromatin structure.

Thank you for suggesting this analysis. We agree that the correlation of histone modification dynamics with allele-specific TF binding would be an interesting direction of research. However, we found this to be beyond the scope of this story for several reasons: first, histone mark abundance is highly dynamic, in contrast to DNA methylation that is highly stable between parents and F1s. Thus, any analyses would have to be performed in the same tissue as the MOA-seq and in the hybrids (which would require repeating all experiments). Additionally, ChIP-seq is a much more complex experiment compared to WGBS or E-methyl-seq. It is highly dependent on antibody quality and would probably require a year of work to achieve good quality data for the 25 hybrids. Furthermore, several histone marks would need to be tested.

The original statement: “SNPs alone explained 45% of the observed differences in MOA coverage, whereas SNPs and DNA methylation combined explained 57%”. Comment: this suggests that DNA methylation does not contribute much to the observed differences in MOA coverage. This result further indicates that histone methylation may be more influential than DNA methylation.

We agree that the way we analyzed the bQTL previously was hard to interpret. We thus modified our strategy and analyzed the contribution of SNPs (or INDELS in our new additional analysis) and methylation separately. We found that 53% of bQTL positions were associated with the genotype allele frequency, 28.9% were associated with methylation level, and the remaining was found in both association analyses. Thus differential methylation was correlated with differential MOA binding in about 47% of our bQTL positions, consistent with the percentages of differential methylation we observe for sites with strong binding bias in single F1s. Differential DNA methylation thus seems to be a strong contributor to haplotype-specific TF binding at functionally relevant sites.

The original statement: “As expected the genome-wide distribution of bQTL was distinct from all SNPs and more closely matched those of TF target sites (Tu et al. 2020) and allele-specific

ZmBZR1 binding sites (Additional file 1, Fig. S8, (Hartwig et al. 2023))”. Comment: This is probably due to linkage disequilibrium.

Thank you for pointing out that this sentence was not clear. We now removed the overlap with TF in general and only concentrated on the fact that our bQTLs, marking putative functional TF binding sites, show a similar distribution towards genes as allele-specific binding sites generated by the gold standard ChIP-seq. This is expected due to the large overlap of ASBs with allele-specific MOA sites but we did not show it for bQTL before in the manuscript and bQTL comprise a more comprehensive set of putative TF binding regions, not just one, so we thought this information might be of interest to the reader. The fact that bQTL distributions differ so strongly from all SNPs was also one of the reasons to opt for matched background SNPs as a control in the VCAP and GWAS hit overlap analyses as we point out below.

The original sentence: “...the causative transposon insertions at ZmCCT9 and ZmCCT10 (Huang et al. 2018)”. Comment: the authors need to cite the relevant article. The transposon insertion at ZmCCT10 was not originally reported in this article.

Thank you for pointing out that we missed this citation. Infact, the bQTL in CCT10 fell below our significance threshold in our new improved analysis with separate methylation and INDEL inclusion (likely due to the increased multiple testing correction). We nevertheless added the citation below to complete the functional description of CCT9/CCT10 regulation.

Citation added:

Qin Yang,a,1 Zhi Li,a,1 Wenqiang Li,b,1 Lixia Ku,c,1 Chao Wang,a Jianrong Ye,a Kun Li,a Ning Yang,b Yipu Li,a Tao Zhong,a Jiansheng Li,a Yanhui Chen,c,2 Jianbing Yan,b,2 Xiaohong Yang,a,2 and Mingliang Xua,2

The original sentence: “We observed haplotype-specific footprints, both at the previously reported 8 bp insertion (Wen et al. 2018) and an additional SNP 29 bp upstream, which coincided with a bQTL”. Comment: are these sequence variations also responsible for the trait difference between B73 and Mo17?

Thank you for this question. We point out that the region was shown to be relevant for trehalose content and expression levels of *TRE1* (Wen et al. 2018). Thus it was shown that variation in this region can influence a molecular phenotype like trehalose levels. B73 and Mo17 are identical in the promoter of *TRE1*, so Mo17 is one of the lines listed as G/G in Fig. 2e. The binding differences towards B73 are found in 6 other F1s.

The original sentence: “whereas F1s with no polymorphism between B73 and NAM showed no difference in the haplotype-specific transcript levels (Fig. 2i-k)”. Comment: This is because the lack of polymorphism failed to distinguish between B73 and NAM, but it does not indicate there is no difference in haplotype-specific transcript levels. For example, the difference in DNA or histone methylation profile resulting from distal transposon insertion must impact haplotype-specific transcript levels.

Thank you for pointing out that this sentence was not formulated precisely. We corrected the sentence by adding “F1s with no polymorphism between B73 and NAM in their promoter/5’UTR”. Indeed we only determined allele-specific transcript abundance for those genes where polymorphisms were present in the transcript (described in the methods section)

and reads were detected for both parental genomes in a mapping mode only allowing reads to map uniquely in the genome with both haplotypes present. Thus, only allele-specific reads were counted. This is why not all 25 lines are displayed in Fig 4k. The four lines (marked pink) that show no difference in the promoter and are displayed had polymorphism in the transcribed region but no significant difference in transcript level.

The original statement: “MOA signals in the TINYB73 and TINYCML333 upstream promoters showed the highest correlation to ZmTINY mRNA levels (Additional File 1, Figure S22)”.
Comment: are these MOA signals responsible for natural variation at TINYB73 and TINYCML333

Thank you for this comment. While an exhaustive analysis on the influence of *TINY* promoter variation on drought susceptibility would take years and is beyond the scope of this study, our transient assays with the CML333 and B73 promoters of *TINY* actually suggest that the promoter region is at least to some extent responsible for the natural variation in mRNA abundance that we observe for *TINY*.

The original statement: “Finally, we highlight the relevance of bQTL loci for understanding phenotypic diversity in maize, demonstrating that haplotype-specific MOA-seq allowed us to capture the majority of additive genetic variation for most tested phenotypes in maize leaves”.
Comment: only the bQTL related to natural variations allowed us to capture the additive genetic variation for most tested phenotypes. However, most bQTL that function in leaf development are not related to genetic variation.

Thank you for pointing this out, we added “genotype-associated” to the sentence.

Fig. 1c, d: These figures are hard to understand and need modification for better clarity.

Thank you for bringing this to our attention, we changed the labelling of these figures to make their meaning clearer and also added more explanation to the legend.

“Fig. 1g,h” In panel g, I do not find much difference in MOA coverage between B73 and Mo17; however, in panel h, MOA coverage in B73 is much higher than that of Mo17.

Thank you for pointing out that the figure needed updating. We now marked the aligned narrow, but significantly differentially bound region in Fig 1g more clearly with dashed orange boxes. We note that during revision we discovered a bug in the software we used to map reads, in which reads without polymorphism were previously counted towards each haplotype instead of the intended random distribution. While this does not affect sites with variation, it affected MOA coverage tracks at non-variant sites. We alerted the software developers, and have redone analyses with the newer version of the software.

Reviewer #3:

Remarks to the Author:

Traditional experimental methods, such as TF ChIP-seq and DNaseI-seq (including ATAC-seq or MOA-seq), have limitations and drawbacks when it comes to find cis-regulatory elements (CREs). On the other hand, computationally inferring CREs from GWAS or other

population genetics-based approaches is prone to false positives, as these methods can identify too many SNPs that are merely linked to the true CRE.

In this study, the authors cleverly used an F1 hybrid approach to overcome the false positive problem. By analyzing the ratio of the two parent's SNPs within the transcription factor (TF) binding sites, they were able to infer whether a particular SNP was actually affecting TF binding, and thus identify the true CREs or at least a small region with multiple SNPs that contain the true CRE. An additional advantage of their approach is that it does not require knowing which specific TF can bind to the identified sites. The authors used MOA-seq, which can broadly identify all TF-protected regions across the genome, without the need to examine each individual TF. Also, the MOA-seq is much easier to performed compared to the DNaseI-seq.

They first tested MOA-seq on the two parental lines, B73 and Mo17, as well as their F1 hybrid. By leveraging their previously published allelic ZmBZR1 binding data for those same lines, they were able to demonstrate that the allelic bias observed in their MOA-seq peaks closely correlated with the allelic patterns of TF binding.

Next, the authors expanded this approach to 25 F1 lines from the NAM population. They reported that the binding QTLs identified through this method could explain 101 out of 143 phenotypic traits in the NAM population, outperforming traditional GWAS (Genome-Wide Association Study) approaches.

Finally, the authors subjected their 25 F1 lines to drought treatment and used the same method to identify cis-regulatory elements (CREs) that are responsive to drought conditions.

While the authors' approach of using F1 hybrids to identify true cis-regulatory elements (CREs) is clever and effective, they should explain to the readers that there is a big limitation. The success of this strategy is dependent on the F1 population. Specifically, the F1 hybrids must have sufficient heterozygosity within the TF binding regions for the method to work effectively. This could be a significant drawback, as other plants, such as arabidopsis, rice and tomato, tend to have much lower levels of heterozygosity compared to maize. So this method is probably only suitable for maize. In addition, this method will miss all the traits that requires a homozygous locus.

Thank you for raising this concern. It is true that lower heterozygosity would mean that more lines will be required to have functional variants of *cis*-elements to be able to identify them. While 25 lines were sufficient for maize, indications from experiments in barley and *Arabidopsis* suggest that with additional lines *cis*-element variants can be identified effectively. At the same time, fewer SNPs allow more precise pinpointing of causative variants with fewer linked variants to consider, a clear advantage of species with less heterozygosity. We discuss these valid points in a statement added to the manuscript: "We note that the high genetic diversity between maize inbred lines allowed us to detect variants with fewer F1 lines. For species with lower diversity, more F1s or the inclusion of more distant interspecies hybrids might be necessary."

It is also worth noting that, even in species with lower diversity, population genetic theory clearly shows that the majority of segregating variation is captured in a relatively small sample

(specifically, the number of polymorphisms observed increases linearly with the log10 of the sample size, so to double the number of polymorphisms captured – in any species – you need to increase sample size by 10X). So while a sample of 25 F1 in tomato may show much less diversity in maize, it nonetheless still captures the majority of the relevant variation in the tomato species.

The trait analysis used phenotypic data from the backcrossed and subsequently selfed NAM mapping population, in which many bQTL variants are homozygous. As such our method allows the identification of the regions which function in homozygous condition. That said, we agree that bQTL can only explain heritability, so trans-effects that exist between inbred lines due to e.g., developmental differences can not be explained in F1 hybrids.

For the title, will it be more accurate to claim that it can "explains phenotypic heritability in maize F1 hybrid"?

Because the phenotypic heritability explained by bQTLs is based on the RIL mapping population not the F1s themselves, it is appropriate to use the statement that bQTLs explain heritability in maize and not just F1 hybrids.

The author frequently used the terms "TF binding footprint" and "cis-regulatory elements (CRE)". Footprint and CRE typically refer to a DNA motif that is bound by the TF, which is ~ 10 bp, while their MOA-seq peak is about 50-100bp? It is better to define it better to help the readers to understand the difference.

We agree and have defined our use of TF footprint more clearly with the added statement: "defined as the area significantly covered by MOA-seq reads". Furthermore, we restricted the use of cis-element to places where we either mention the potential of cis-element presences or where we directly talk about motifs that were affected and displayed differential binding.

The penultimate result section compares their method with others. I think this is an important part. But it is quite hard to read. Background SNP account for more genetic variation? Why? How do they match the distance to nearest gene? Are they in the open chromatin? Using whole-genome's SNP can't get similar result? Will it be the same to use allelic SNP in ATAC-seq peaks? I hope the author can improve this part and let the reader understand what is the pros and cons of their method.

Thank you for pointing out that parts of this paragraph were unclear. We now added that the background SNPs match bQTL in their allele frequency and distance to genes more early on in the paragraph and explain that this was done to account for the difference in distribution of bQTL in relation to genes compared to all SNPs. The "rest of the genome" fraction is in fact sampled from all SNPs excluding bQTL and background SNPs; we also made this more clear now. We expected the bgSNPs to perform better in VCAP than the rest of the genome because they are closer to genes (because they match the same distribution as bQTL) and thus are more likely located to regions relevant for gene regulation as shown in Rodgers-Melnick et al. 2016. We hope this now helps to convey our message that bQTL outperform both background and "rest of the genome" SNPs, while the background SNPs outperform the rest of the genome in most cases.

Thank you also for raising the question about ATAC-seq as an alternative to MOA-seq. We decided to address this question in the conclusion and added a paragraph there. While we think that ATAC-seq or even ChIP-seq for single TFs can be used in combination with F1 hybrids to obtain similar results, previous studies have shown that MOA-seq detects additional sites (probably due to the larger size of the Tn5-dimer used in ATAC) and will thus probably yield a more comprehensive data set.

The method section lacks details to reproduce the analysis. I understand the space limit but they can and should put a detail method into the supplementary information. Also they should put their raw data to NCBI SRA and processed data to GEO. Currently, the link they provide are not accessible. So I can't check the quality of the data.

We have a reviewer link for the SRA data that can be viewed and we added two more github repositories describing the allele-specific analyses in detail. We also added the transformed data (Peaks, bigwig tracks) to GEO but awaiting the ID release. We also added more description at several places in the methods section.

Reviewer link sra:

<https://dataview.ncbi.nlm.nih.gov/object/PRJNA1101486?reviewer=cb1oa1gh0h8gem4kbvpt57jtfi>